# PrunedLoRA: Robust Gradient-Based structured pruning for Low-rank Adaptation in Fine-tuning

## Abstract

Low-rank adaptation (LoRA) has become a widely used paradigm for parameter-efficient fine-tuning of large language models, yet its empirical performance often lags behind full fine-tuning due to its low-rank constraint. Within the context of LoRA, a key open question is how to obtain expressive low-rank adapters from over-parameterized spaces. We propose *PrunedLoRA*, a new framework that leverages structured pruning to obtain highly representative low-rank adapters from an over-parameterized initialization. Unlike prior approaches that impose a fixed low-rank budget, *PrunedLoRA* dynamically prunes less important components during fine-tuning and prevents their reactivation, enabling flexible and adaptive rank allocation. For structured pruning, by minimizing the pruning error for overall loss, we provide fine-grained pruning and recovery updates in a gradient-based pruning strategy with grounded interpretation. We provide the first theoretical analysis of the robustness of structured pruning and provably show that under the impact of weight perturbation, gradient-based pruning is more robust than activation-based pruning with respect to overall loss. Empirically, *PrunedLoRA* consistently outperforms LoRA and its variants across supervised fine-tuning tasks in mathematical reasoning, code generation, and natural language understanding, and it also demonstrates advantages over existing structured pruning methods across diverse sparsity levels.

## 1 Introduction

Low-rank adaptation (LoRA) (Hu et al., 2022) and its variant (Zhang et al., 2023; Liu et al., 2024; Hayou et al., 2024) have emerged as a prominent class of parameter-efficient fine-tuning (PEFT) methods for large-scale foundation models (Sidahmed et al., 2024; Luo et al., 2023; Zhao et al., 2025). By injecting trainable low-rank matrices into the pre-trained model, LoRA enables efficient fine-tuning with minimal training overhead and no additional inference latency. Despite its efficiency, LoRA often lags behind full fine-tuning (FFT) in practical performance. Existing attempts to bridge this gap fall into two categories. The first line of work strictly follows LoRA's memory constraint, so exploring over the full parameter space is inadmissible (Hayou et al., 2024; Yen et al., 2024; Kalajdzievski, 2023; Chen et al., 2025). Learning within the low-rank space is always difficult to utilize the powerful representation of FFT (Zhang et al., 2025; Hao et al., 2024). The second line of work enables full-parameter learning (Zhao et al., 2024; He et al., 2024; Liao et al., 2024) through projection techniques to compress and decompress gradients and weights. While these over-parameterized methods improve performance, they ultimately output fine-tuned full models rather than preserving a shared base model with lightweight, task-specific low-rank adapters. As a result, for the inference period, these approaches with full-parameter learning are less efficient, since each task requires storing a full model. In contrast, if we obtain low-rank adapters for different tasks, inference time and memory cost can be significantly reduced (Yang et al., 2024; Liao et al., 2025; Feng et al., 2024). Therefore, the key question remains open: *how can we find highly representative low-rank adapters from an over-parameterized setting for better performance?*

Empirically, we observe that increasing the rank of LoRA improves performance, in some cases approaching that of FFT (see Fig. 1 in Subsection 3.1), a trend also reported in prior work (Wang et al., 2024b; Hu et al., 2022). This suggests that LoRA with a larger rank has sufficient represen-

tational capacity. Motivated by this observation, we consider initializing LoRA with a larger rank to ensure sufficient representational capacity, and then reducing the size of the model during fine-tuning to obtain a lightweight low-rank adapter. This strategy preserves the expressive power of an over-parameterized initialization while maintaining efficiency.

To realize this idea, we next turn to structured pruning (LeCun et al., 1989; Hassibi et al., 1993; Denil et al., 2013; Zhu & Gupta, 2017), a principled approach for reducing the model size by removing entire sub-components, such as rows or columns, from the model's weight matrices. Two main categories of structured pruning have been widely studied: gradient-based methods (Molchanov et al., 2019a; Yang et al., 2023b; Ma et al., 2023) and activation-based methods (Frantar & Alistarh, 2023; Kurtić et al., 2023; Zhao et al., 2022). Empirical evidence (e.g., Nonnenmacher et al. (2021)) suggests that gradient-based approaches focus more on global information and would be more stable for overall loss under weight perturbations. However, from a theoretical perspective, a clear comparison between these two classes of methods, particularly regarding how weight perturbations affect the overall loss, remains largely unexplored. To further mitigate the influence of pruning, Frantar & Alistarh (2023); Kurtić et al. (2023); Singh & Alistarh (2020) proposes updating weights after pruning, inspired by *Optimal Brain Surgeon* (Hassibi & Stork, 1992). While these approaches investigate how to scale second-order methods to deep neural networks, they, as the original work Hassibi & Stork (1992), leave open a deeper understanding of the pruning metric, known as "saliency" term in *Optimal Brain Surgeon*.

In this work, with the goal of narrowing the gap between LoRA and full fine-tuning, we propose *PrunedLoRA*, enabling full-parameter learning while dynamically pruning the initial weights from an over-parameterized space. Unlike existing methods focusing on a fixed low-rank budget, *PrunedLoRA* enjoys the freedom of learning from over-parameterized spaces while converging to lightweight low-rank adapters for training and inference efficiency. For the theoretical analysis of structured pruning, we consider a toy model of self-attention (Vaswani et al., 2017) and provably show that gradient-based pruning is more robust to weight perturbations in terms of overall loss than activation-based pruning approaches. We further show that this intuition extends to broader contexts. In addition, we provide a fine-grained analysis of pruning selection and weight update for weight matrices in a second-order gradient-based pruning strategy, which deepens the understanding of the pruning metric (the "saliency" term in Eq. 5 of Hassibi & Stork (1992)) in the class of second-order pruning methods.

We summarize our contribution as follows:

- We propose *PrunedLoRA*, a new framework that identifies highly representative low-rank adapters by structured pruning from an over-parameterized initialization with more representation capacity while retaining efficiency. Unlike prior approaches with a fixed low-rank budget, *PrunedLoRA* only enforces the low-rank constraint at the end of fine-tuning, enabling flexible and adaptive rank allocation during fine-tuning.

- We establish the first theoretical analysis of the robustness of two major structured pruning approaches for large language models. Using a toy self-attention model, we prove that gradient-based pruning is more robust to weight perturbations in terms of overall loss than activation-based pruning, and we also show that this intuition extends to broader settings.

- We conduct extensive experiments across supervised fine-tuning tasks spanning mathematical reasoning, code generation, and natural language understanding, showing that *PrunedLoRA* can further narrow the gap between LoRA and FFT. Across different sparsity levels from $50\%$ to $93\%$ and across various pruning tasks (including both dynamic and one-shot pruning), our method consistently outperforms existing structured pruning methods.

## 2 RELATED WORK

**Low-rank adaptation (LoRA)** has been extensively investigated in foundation models (Chen et al., 2023; Wei et al., 2024a; Bai et al., 2024), with numerous variants and enhancements proposed (Meng et al., 2024; Hayou et al., 2024; Wang et al., 2024a). Hu et al. (2022) assumes that the fine-tuning update can be effectively captured in a low-rank subspace. Specifically, for a pre-trained model with weight matrix $\boldsymbol{W}_0 \in \mathbb{R}^{m \times n}$, LoRA reparameterizes the weight update $\Delta \boldsymbol{W}$ via a low-rank decomposition as $\boldsymbol{W}_0 + \Delta \boldsymbol{W} = \boldsymbol{W}_0 + s\boldsymbol{B}\boldsymbol{A}$, where $\boldsymbol{B} \in \mathbb{R}^{m \times r}$, $\boldsymbol{A} \in \mathbb{R}^{r \times n}$ and $s = \frac{\alpha}{r}$ is a scaling factor. Here, $r \ll min(m, n)$ is the rank of the update. AdaLoRA (Zhang et al., 2023)

dynamically allocates the parameter budget by assigning more capacity to task-critical modules, but remains constrained within a limited subspace and does not fully explore the parameter space as in full fine-tuning. LoRA-Prune (Zhang et al., 2024) leverages gradients from LoRA modules rather than the entire model to prune the whole model, which differs from our goal and leads to substantial performance degradation. In contrast, we only prune the trainable parameters to produce representative low-rank adaptations at the end.

**Compression of Large Language Model (LLM)** has gained a lot of attention and has been widely applied for parameter efficiency and reducing the latency (Lan et al., 2019; Sun et al., 2020b). To compress the language model, previous works can be divided into several categories: network pruning (Kurtic et al., 2022; Xu et al., 2021; Liu et al., 2021; Guo et al., 2019), knowledge distillation (Sun et al., 2019; 2020a; Pan et al., 2021), quantization (Yao et al., 2022; Bai et al., 2021; Zafrir et al., 2019) and other techniques, like early exit (Xin et al., 2020). In this work, we focus on structurally network pruning (Li et al., 2017) to remove the entire filter from the neural network, whose approaches can be mainly categorized into two lines: activation-based pruning and gradient-based pruning. For the activation-based pruning Dubey et al. (2018); Hu et al. (2016), it explores structured pruning based on activation statistics of neuron/filter output. If we aim to prune the weight matrix $\boldsymbol{W}$, many activation-based strategies (Frantar & Alistarh, 2023; Kurtić et al., 2023; Xie et al., 2024; Wei et al., 2024b) focus on the following optimization problem

$$argmin_{\widehat{\boldsymbol{W}} \in \mathbb{R}^{m \times n}} \left\| \widehat{\boldsymbol{W}} \boldsymbol{X} - \boldsymbol{W} \boldsymbol{X} \right\|^2 \quad s.t. \quad \widehat{\boldsymbol{W}} \in \mathcal{C}, \tag{1}$$

where $\mathcal{C}$ is a certain sparse structure. Inspired by Optimal Brain Surgeon (Hassibi & Stork, 1992), finding the optimal $\widehat{\boldsymbol{W}}$ in (1) takes two steps: find the optimal pruning column first and update the unpruned column (Tang et al., 2025; Kurtić et al., 2023; Li et al., 2025a). For gradient-based strategies, by allowing access to the gradient of the overall loss, to measure the importance of $i$-th column in $\widehat{\boldsymbol{W}}$, Zhang et al. (2023); Yang et al. (2023b) estimate the change in loss $\mathcal{L}$ once pruning the $i$-th column:

$$I_{\boldsymbol{W}_i} = |\Delta \mathcal{L}_{\boldsymbol{W}_i}| = |\mathcal{L}_{\boldsymbol{W}_i} - \mathcal{L}_{\boldsymbol{W}_i=0}|. \tag{2}$$

Here, computing the important score can help to find the pruned column, but it keeps the unpruned weight unchanged, without compensating for the influence of pruning. Thus, for a weight matrix, *how to minimize the influence of pruning in gradient-based methods* is important.

# 3 METHODS

## 3.1 MOTIVATION

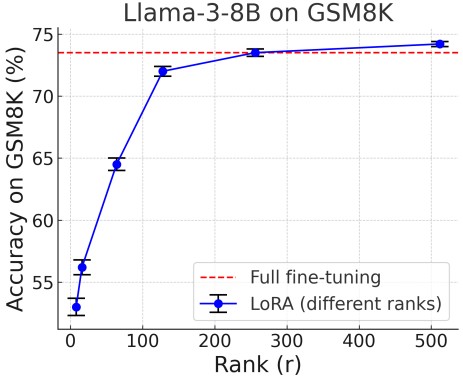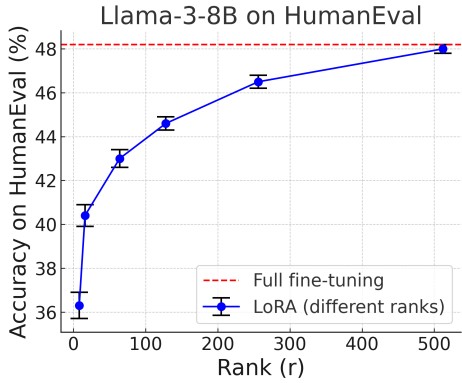

Figure 1: Performance of standard LoRA (Hu et al., 2022) on GSM8K (Cobbe et al., 2021) and HumanEval (Chen et al., 2021) with different ranks compared to full fine-tuning. Note that the method of full fine-tuning does not involve the initial rank, and we draw a red line here solely for comparison.

**Motivation 1: Higher rank results in better performance.** As illustrated in Figure 1, employing higher ranks in LoRA consistently leads to improved empirical performance on both GSM8K and

HumanEval (see Sec. 4 for details). Notably, as the rank increases, the performance gradually converges toward that of full fine-tuning. This observation motivates our approach: rather than fixing LoRA to a small rank at the outset, we initialize with a sufficiently large rank—providing a number of trainable parameters close to full fine-tuning—and then progressively prune it to a smaller rank. Such a strategy may preserve most of the performance gains in over-parameterized settings while ultimately producing a memory-efficient low-rank adaptation.

**Motivation 2: $A$ and $B$ in LoRA control the low-rank spaces.** For the sub matrices, $A \in \mathbb{R}^{r \times n}$ and $B \in \mathbb{R}^{m \times r}$, we observe that the columns of $B$ correspond to the column space of the original update $\Delta W$, while the rows of $A$ represent the row space (Yu et al., 2025). Therefore, they can capture the row-wise and column-wise correlation separately. As we will discuss in the next section, pruning on sub-modules instead of the full matrix reduces the computational cost and simplifies the second-order structured pruning significantly.

### 3.2 THE ROBUSTNESS OF GRADIENT-BASED STRUCTURED PRUNING

**Activation-based *v.s.* Gradient-based structured pruning.** Pruning induces perturbations to the weights across layers of large language models, which in turn modifies the overall loss and may lead to a deterioration of empirical performance (Frantar & Alistarh, 2023; Yang et al., 2023a). Within the context of structured pruning (Liu et al., 2017; Molchanov et al., 2019b; Fan et al., 2019), activation-based solving Problem (1) and gradient-based pruning using important scores in (2) are two main lines of approaches to find the optimal pruned structure. Intuitively, gradient-based methods focus more on the global correlation (Nonnenmacher et al., 2021), so they shall be more robust for the overall loss under the influence of weight perturbation. However, no theoretical analysis provably shows the insight. Here, we analyze the influence of different pruning strategies on the overall loss. We provide formal analysis and general discussion in Appendix B.

**Proposition 1 (Unofficial Statement)** *Suppose that, under activation-based and gradient-based pruning strategies, each module in a single attention module satisfies a given perturbation error. The error in the loss function would be linear w.r.t. perturbation error under different pruning strategies, but the error of activation-based methods depends on the magnitude of each module.*

Proposition 1 reveals that the activation-based methods introduce a higher infatuation for the overall loss. It is consistent with the insight that activation-based methods cannot indicate the influence of weight change for global correlation (Das et al., 2023). Within the context of gradient-based pruning strategies, we formulate our problem on pruning the columns of a full weight matrix first. It would be interpreted as pruning the columns of matrix $B$ (or the rows of matrix $A$) alone.

**Problem formulation.** Our approach starts from the idea of applying a structured compression layer-wise, in a way that allows the layers to preserve most of their output characteristics. This setup is popular in the post-training quantization and unstructured pruning literature (Frantar & Alistarh, 2023; Tang et al., 2025; Wu et al., 2024), and can be implemented as follows. In the fine-tuning period, the gradient is non-trivial as it helps the fine-tuned model align with the down-task data. Therefore, our setup is different from the literature in gradient-based pruning (Singh & Alistarh, 2020; Kurtic et al., 2022). We consider the perturbation of a single weight matrix $W \in \mathbb{R}^{m \times n}$ in a large language model. The pruned matrix is denoted as $W + \delta$, where the perturbation $\delta \in \mathbb{R}^{m \times n}$ corresponds to pruning the same weight indices across all rows, i.e., entire columns are removed. The update $\delta \in \mathbb{R}^{m \times n}$ is subject to the constraint that

$$\delta_{:,\mathcal{M}_s} = -W_{:,\mathcal{M}_s}. \tag{3}$$

Here, $\mathcal{M}_s$ denotes the pruning mask that specifies the pruned column indices with sparsity s. Expanding the overall loss of the pruned model with weight matrix $W + \delta$ around $W$ yields

$$\mathcal{L}(W + \delta) \approx \mathcal{L}(W) + \langle \nabla_W \mathcal{L}(W), \delta \rangle + \frac{1}{2} tr(vec(\delta)^\top \mathbf{H} \, vec(\delta)), \tag{4}$$

which corresponds to the matrix-form second-order Taylor expansion, where $vec(\delta)$ denotes the vectorization of the perturbation matrix. Noticeably, the Hessian matrix is $\mathbf{H} \in \mathbb{R}^{mn \times mn}$, so the memory cost and the computational cost are extremely huge. To address the challenge, many existing methods propose to impose structural assumptions for the Hessian matrix $\mathbf{H}$, such as diagonal or block-diagonal approximation (Zhang et al., 2017; Hassibi & Stork, 1992) and empirical

Fisher (Cho et al., 2015; Singh & Alistarh, 2020). With the goal of selecting columns in (3), it is critical to preserve the correlation among the columns of the weight matrix. Thus, with the standard assumption of row independence in Kurtic et al. (2022); Frantar & Alistarh (2023), as a common technique for approximating the Hessian using gradients, we can approximate (4) by

$$\mathcal{L}(\boldsymbol{W} + \boldsymbol{\delta}) \approx \mathcal{L}(\boldsymbol{W}) + \langle \nabla_{\boldsymbol{W}}\mathcal{L}(\boldsymbol{W}), \boldsymbol{\delta} \rangle + \frac{1}{2}tr(\boldsymbol{\delta}^{\top}\widehat{\boldsymbol{H}}\boldsymbol{\delta}), \tag{5}$$

where $\widehat{\boldsymbol{H}} = (\nabla_{\boldsymbol{W}}\mathcal{L}(\boldsymbol{W}))^{T}\nabla_{\boldsymbol{W}}\mathcal{L}(\boldsymbol{W}) \in \mathbb{R}^{n \times n}$. Then, combining the pruned structure (3) with the analysis of perturbation in $\boldsymbol{W}$, it yields the optimal pruning selection and weight update by solving the following problem:

$$\mathcal{M}_s, \boldsymbol{\delta} = argmin_{\mathcal{M}_s, \boldsymbol{\delta}} \langle \nabla_{\boldsymbol{W}}\mathcal{L}, \boldsymbol{\delta} \rangle + \frac{1}{2}tr(\boldsymbol{\delta}\widehat{\boldsymbol{H}}\boldsymbol{\delta}^{\top})$$
$$s.t. \quad \boldsymbol{\delta}_{:,\mathcal{M}_s} = -\boldsymbol{W}_{:,\mathcal{M}_s}. \tag{6}$$

Here, for simplicity, we denote $\nabla_{\boldsymbol{W}}\mathcal{L}(\boldsymbol{W})$ as $\nabla_{\boldsymbol{W}}\mathcal{L}$. The optimal solution of $\boldsymbol{\delta}$ in ( 6) is

$$\boldsymbol{\delta} = -\nabla_{\boldsymbol{W}}\mathcal{L}\,\widehat{\boldsymbol{H}}^{-1} - \boldsymbol{W}_{:,\mathcal{M}_s}\left((\widehat{\boldsymbol{H}}^{-1})_{\mathcal{M}_s,\mathcal{M}_s}\right)^{-1}(\widehat{\boldsymbol{H}}^{-1})_{\mathcal{M}_s,:}$$
$$+ (\nabla_{\boldsymbol{W}}\mathcal{L}\widehat{\boldsymbol{H}}^{-1})_{:,\mathcal{M}_s}\left((\widehat{\boldsymbol{H}}^{-1})_{\mathcal{M}_s,\mathcal{M}_s}\right)^{-1}(\widehat{\boldsymbol{H}}^{-1})_{\mathcal{M}_s,:}. \tag{7}$$

**Interpretation for Algorithm Design.** Let us further analyze the update $\boldsymbol{\delta}$ in (7). The first term in $\boldsymbol{\delta}$ is a second-order Newton step. If there is no sparse masking, it would be the optimal update utilizing second-order momentum. As $P_{\mathcal{M}_s}\boldsymbol{\delta}$ will only leave the second term in (7), which is a projection correction to ensure the pruned weights remain zero. Interestingly, it is dependent on the current weight $\boldsymbol{W}$ and the mask $\mathcal{M}_s$ but independent of the gradient $\nabla_{\boldsymbol{W}}\mathcal{L}$. The third term in (7) provides a dual variable compensation that projects the unconstrained Newton step into the feasible region. Once we get the closed-form solution of $\boldsymbol{\delta}$ in (7), the pruning problem in (14) is

$$min_{\mathcal{M}_s} \, tr\left((\mathbf{W} - \nabla_{\boldsymbol{W}}\mathcal{L}\,\mathbf{H}^{-1})_{:,\mathcal{M}_s}\,\left((\mathbf{H}^{-1})_{\mathcal{M}_s,\mathcal{M}_s}\right)^{-1}\,(\mathbf{W} - \nabla_{\boldsymbol{W}}\mathcal{L}\,\mathbf{H}^{-1})_{:,\mathcal{M}_s}^{\top}\right). \tag{8}$$

Here, the pruning problem in (8) is closely related to the "saliency" term in (Hassibi & Stork, 1992). With the analysis of matrix weight, we provide *an explicit interpretation* for second-order pruning strategies: *we select the pruning mask that removes the columns whose post-update (second-order Newton update) values are least important under the Hessian-weighted quadratic metric.* Existing methods deriving from Optimal Brain Surgeon can not provide a grounded interpretation from the "saliency" term, as most of them focus on the specific problems such as (1) (Frantar & Alistarh, 2023; Kurtić et al., 2023) or only analyze the one-dimensional weight vectors (Das et al., 2023; Singh & Alistarh, 2020; Kurtic et al., 2022). Therefore, our analysis enriches the understanding of the class of second-order pruning methods.

We summarize our solution in Algorithm 2 and we present a schematic illustration of the workflow in the left of Figure 2. In each pruning step, the pruning indices are determined by the gradient and the estimated Hessian.

### 3.3 PRUNEDLORA

In this part, we propose our structured pruning strategy, termed *PrunedLoRA*. Inspired by *Motivation 1*, we dynamically prune adapters $\boldsymbol{A}$ and $\boldsymbol{B}$ from high-parameter spaces.

Different from prior work such as AdaLoRA (Zhang et al., 2023), which enforces an average rank budget and dynamically selects ranks from a small predefined set. It always restricts the rank of the updated weight in low-rank spaces. Besides, structurally pruning the columns and rows of a full weight matrix causes high computational overhead, as we highlight in Eq. (4). However, with *Motivation 2*, we can efficiently detect the row-wise and column-wise correlation by pruning the low-rank spaces of $\boldsymbol{A}$ and $\boldsymbol{B}$ together. With the goal of reducing the rank of the matrix, structured pruning of the decomposed sub-modules would be more efficient.

With the standard argument in Sec 3.2, the pruning problem for low-rank adaptation $\boldsymbol{A}$ and $\boldsymbol{B}$ is

$$argmin_{\mathcal{M}_s, \boldsymbol{\delta}_{\boldsymbol{A}}, \boldsymbol{\delta}_{\boldsymbol{B}}} \langle \nabla_{\boldsymbol{A}}\mathcal{L}, \boldsymbol{\delta}_{\boldsymbol{A}} \rangle + \frac{1}{2}tr(\boldsymbol{\delta}_{\boldsymbol{A}}^{\top}\widehat{\boldsymbol{H}}_{\boldsymbol{B}}\boldsymbol{\delta}_{\boldsymbol{A}}) + \langle \nabla_{\boldsymbol{B}}\mathcal{L}, \boldsymbol{\delta}_{\boldsymbol{B}} \rangle + \frac{1}{2}tr(\boldsymbol{\delta}_{\boldsymbol{B}}\widehat{\boldsymbol{H}}_{\boldsymbol{B}}\boldsymbol{\delta}_{\boldsymbol{B}^{\top}})$$
$$s.t. \quad (\boldsymbol{\delta}_{\boldsymbol{B}})_{:,\mathcal{M}_s} = -\boldsymbol{B}_{:,\mathcal{M}_s}, \quad (\boldsymbol{\delta}_{\boldsymbol{A}})_{\mathcal{M}_s,:} = -\boldsymbol{A}_{\mathcal{M}_s,:}. \tag{9}$$

---

**Algorithm 1 PrunedLoRA**: structured pruning for Low-rank Adapters from over-parameterized spaces. We prune LoRA matrices $(A, B)$ with column sparsity $s$ on $B$ (and corresponding row sparsity s on $A$) given gradients $(\nabla_A \mathcal{L}, \nabla_B \mathcal{L})$ and Hessian estimates $(\widehat{H}_A, \widehat{H}_B)$.

---

1: **Step 1: Search pruning mask.**

$$\arg \min_{\mathcal{M}_s} \; \mathrm{tr}\left( \widetilde{B}_{:,\mathcal{M}_s} ((\widehat{H}_B^{-1})_{\mathcal{M}_s,\mathcal{M}_s})^{-1} \widetilde{B}_{:,\mathcal{M}_s}^{\top} \right) + \mathrm{tr}\left( \widetilde{A}_{\mathcal{M}_s,:}^{\top} ((\widehat{H}_A^{-1})_{\mathcal{M}_s,\mathcal{M}_s})^{-1} \widetilde{A}_{\mathcal{M}_s,:} \right),$$

where $\widetilde{A} = A - \widehat{H}_A^{-1} \nabla_A \mathcal{L}$, $\widetilde{B} = B - \nabla_B \mathcal{L} \, \widehat{H}_B^{-1}$.

2: **Step 2: Compute optimal updates.**
3: Given $\mathcal{M}_s$, compute

$$\delta_B = -\nabla_B \mathcal{L} \, \widehat{H}_B^{-1} - \widetilde{B}_{:,\mathcal{M}_s} ((\widehat{H}_B^{-1})_{\mathcal{M}_s,\mathcal{M}_s})^{-1} (\widehat{H}_B^{-1})_{\mathcal{M}_s,:},$$

$$\delta_A = -\widehat{H}_A^{-1} \nabla_A \mathcal{L} - (\widehat{H}_A^{-1})_{:,\mathcal{M}_s} ((\widehat{H}_A^{-1})_{\mathcal{M}_s,\mathcal{M}_s})^{-1} \widetilde{A}_{\mathcal{M}_s,:}.$$

4: Set $A \leftarrow A + \delta_A$, $B \leftarrow B + \delta_B$.
5: **Step 3: Update LoRA adapters with standard optimizer in fine-tuning.**
6: **Step 4: Iterate or finalize.**
7: If multi-round pruning is desired, repeat Steps 1–3 until the target rank is reached. Otherwise, output $(A, B)$.

---

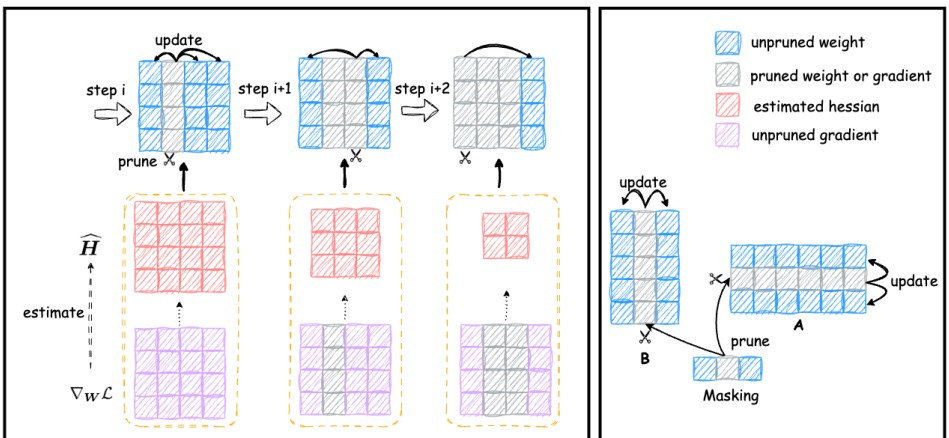

Figure 2: **Left:** schematic of the dynamic pruning process, where the gradient and estimated Hessian will determine pruned columns and update as shown in Algorithm 2. **Right:** design of *PrunedLoRA*, where both adapter matrices $A$ and $B$ are jointly pruned under a masking scheme.

Here, the mask $\mathcal{M}_s$ simultaneously controls the column sparsity of $B$ and the row sparsity of $A$. Consequently, the Hessian estimates $\widehat{H}_A$ and $\widehat{H}_B$ are computed with different purposes: to capture the column-wise correlations of $B$ and the row-wise correlations of $A$, respectively. Following the standard derivation in Sec 3.2, our pruning strategy for reducing high-rank matrices $A$ and $B$ to a low-rank adaptation begins by determining the optimal pruning mask via

$$argmin_{\mathcal{M}_s} \, \mathrm{tr}\left( \widetilde{B}_{:,\mathcal{M}_s} ((\widehat{H}_B)_{\mathcal{M}_s,\mathcal{M}_s}^{-1})^{-1} \widetilde{B}_{:,\mathcal{M}_s}^{T} \right) + \mathrm{tr}\left( \widetilde{A}_{\mathcal{M}_s,:}^{T} ((\widehat{H}_A)_{\mathcal{M}_s,\mathcal{M}_s}^{-1})^{-1} \widetilde{A}_{\mathcal{M}_s,:} \right), \quad (10)$$

where $\widetilde{A} = A - (\widehat{H}_A)^{-1} \nabla_A \mathcal{L}$, $\widetilde{B} = B - \nabla_B \mathcal{L} (\widehat{H}_B)^{-1}$. After selecting the pruning indices, we update $A$ and $B$ as (11) to minimize the perturbation error in the loss.

$$\delta_B = -\nabla_B \mathcal{L} \, \widehat{H}_B^{-1} - \widetilde{B}_{:,\mathcal{M}_s} ((\widehat{H}_B^{-1})_{\mathcal{M}_s,\mathcal{M}_s})^{-1} (\widehat{H}_B^{-1})_{\mathcal{M}_s,:}$$
$$\delta_A = -\widehat{H}_A^{-1} \nabla_A \mathcal{L} - (\widehat{H}_A^{-1})_{:,\mathcal{M}_s} ((\widehat{H}_A^{-1})_{\mathcal{M}_s,\mathcal{M}_s})^{-1} \widetilde{A}_{\mathcal{M}_s,:}. \quad (11)$$

**Complexity.** For *PrunedLoRA*, the pruning procedure begins with an initial rank much smaller than $\min\{m, n\}$ and progressively reduces the rank until reaching the target level. Since pruning is performed only for a limited number of steps, the additional cost introduced by the pruning operations remains moderate. Moreover, even for the initialization of LoRA with high rank, the computational complexity of pruning is $\mathcal{O}(r^3)$ and much smaller than that of matrix multiplication, i.e., $\mathcal{O}(\max\{m^2 r, n^2 r\})$. The computational time of pruning is mild. For instance, we isolate and measure the wall-clock time required solely for the pruning procedure with initial rank $r = 512$ on the GSM8K datasets using the pretrained Llama3-8b base model. The time consuming for pruning is 13 mins (6.40% in overall training time), demonstrating that the additional cost introduced by our second-order pruning strategy is modest in practice. Consequently, our method maintains a computational cost comparable to that of existing low-rank adaptation approaches (Yen et al., 2024; Yu et al., 2025).

## 4 EXPERIMENT

In this section, we present extensive experiments to evaluate the effectiveness of *PrunedLoRA* across various tasks and models. With different levels of pruning sparsity, we assess its capabilities on supervised fine-tuning tasks in dialogue generation, mathematical reasoning, code generation using the Llama-3-8B model (Grattafiori et al., 2024), and natural language understanding on a T5-based model covered in Sec 4.1. Then we conduct ablation studies for the hyperparameters, pruning schedules, and more pruning baselines in Sec 4.2 and Appendix C.5. In addition to conducting structured pruning to obtain low-rank adaptation in fine-tuning, one-shot pruning for compressing a pretrained model is crucial in the pre-LLM era (Sun et al., 2023) as well, but most of the work (Han et al., 2015; Sun et al., 2023; Frantar & Alistarh, 2023) is activation-based methods without awareness of the influence of weight perturbation on the overall loss function. We provide a simple gradient-based method as well in Appendix D without weight update. It supports the effectiveness of gradient-based pruning strategies for eliminating the impact of weight perturbation.

**Baselines.** We compare *PrunedLoRA* with several representative fine-tuning paradigms to demonstrate its effectiveness. The first baseline is *Full Fine-Tuning*, where all parameters are updated. While this approach typically achieves the best performance, it is computationally expensive and offers no gains in inference efficiency. A widely adopted alternative is vanilla *LoRA* (Hu et al., 2022), which reparameterizes the updates through low-rank adapters $A$ and $B$, initialized with Gaussian noise for $A$ and zeros for $B$. We further consider two prominent LoRA variants that modify the low-rank structure: DoRA (Liu et al., 2024), which enhances representational capacity via learnable magnitude scaling, and AdaLoRA (Zhang et al., 2023), which adaptively prunes and reallocates ranks based on singular value decomposition (SVD) to better capture parameter importance under a fixed budget. These variants constitute the most widely used structural extensions of LoRA. For a comprehensive and fair comparison, we also compare *PrunedLoRA* with other variants of LoRA (e.g LoRA+ (Hayou et al., 2024), LoRA-GA (Wang et al., 2024a), LoRA-Pro (Wang et al., 2024b), AltLoRA(Yu et al., 2025), HiRA (Huang et al., 2025), MoRA (Jiang et al., 2024) and ABBA Singhal et al. (2025)) in the Appendix C.4.

In addition to fine-tuning baselines, we also compare against existing structured pruning approaches for low-rank adaptation. Gradient-based pruning includes our method, which jointly optimizes parameter updates and pruning structure, as well as the widely used importance-score pruning strategy (Eq. 2) employed in LLM-Pruner (Ma et al., 2023). Activation-based pruning determines the pruning structure based on input activation statistics (Eq. 1), as exemplified by ZipLM (Kurtić et al., 2023) and SparseGPT (Frantar & Alistarh, 2023). We further include comparisons with other classical pruning strategies (Sun et al., 2023; Han et al., 2015), along with one-shot pruning, which are reported in Appendix C.5.

### 4.1 EXPERIMENTS ON SUPERVISED FINE-TUNING

**Implementation Details.** To ensure a fair comparison, we align our experimental setup with the literature (Wang et al., 2024a;b; Yu et al., 2025). We fine-tune the model using the AdamW optimizer with hyperparameters $\beta_1 = 0.9, \beta_2 = 0.999$, and weight decay set to 0. We implement a cosine learning rate schedule with a warm-up ratio of 0.03. LoRA is applied to all linear modules, excluding the embedding layer and normalization layer. All experiments are conducted on NVIDIA H100

GPUs. To obtain a reliable estimate of model performance, we perform three runs with different random seeds and report the average and standard deviation of the results.

For dialogue generation, mathematical reasoning, and code generation tasks, we set the target LoRA rank $r \in \{8, 64\}$ and search the scaling factor $\alpha$ over $\{r/2, r, 2r\}$. For structural pruning methods, the initial ranks are set to 64 and 128 as default, respectively, to ensure a sufficiently expressive pre-pruning space. For the learning rate, we perform a grid search over $\{2 \times 10^{-4}, 5 \times 10^{-5}, 2 \times 10^{-5}\}$. This hyperparameter range fully covers the settings used in prior work (Wang et al., 2024a;b; Yu et al., 2025). We use a sequence length of 1024 and a macro batch size of 32.

For natural language understanding tasks, we fine-tune the T5-base model (Raffel et al., 2020) with learning rate of $1e{-}4$ and target LoRA rank 8, using a sequence length of 128 and a batch size of 32. For structural pruning methods, we set the initial LoRA rank to 64. For DoRA, we adopt a learning rate of $2 \times 10^{-4}$, while for AdaLoRA, we follow prior work and use $5 \times 10^{-4}$. This configuration exactly matches the experimental setups used in Wang et al. (2024a;b); Yu et al. (2025).

**Results on Natural Language Generation.** Following the configuration used in (Wang et al., 2024a;b; Yu et al., 2025), we evaluate the performance of PrunedLoRA on large language models, focusing on dialogue generation, mathematical reasoning and code generation capabilities. We provide the detailed descirption about the three tasks in Appendix C.1.

| Method | Target Rank | MT-Bench ↑ | GSM8K ↑ | HumanEval ↑ |
|---|---|---|---|---|
| PreTrain | – | $5.89 \pm 0.04$ | $51.34 \pm 1.38$ | $34.21 \pm 0.23$ |
| Full FT | – | $\mathbf{6.31 \pm 0.03}$ | $\underline{73.48 \pm 0.42}$ | $\underline{48.28 \pm 0.03}$ |
| LoRA | 8 | $6.01 \pm 0.05$ | $65.27 \pm 0.13$ | $39.23 \pm 0.78$ |
|  | 64 | $6.19 \pm 0.03$ | $69.21 \pm 0.36$ | $42.88 \pm 0.34$ |
| DoRA | 8 | $6.07 \pm 0.02$ | $67.08 \pm 0.31$ | $41.28 \pm 0.39$ |
|  | 64 | $6.23 \pm 0.03$ | $70.43 \pm 0.21$ | $43.32 \pm 0.29$ |
| AdaLoRA | 8 | $6.08 \pm 0.03$ | $71.24 \pm 1.32$ | $41.88 \pm 1.15$ |
|  | 64 | $6.12 \pm 0.08$ | $71.45 \pm 1.37$ | $42.34 \pm 1.41$ |
| SparseGPT | 8 (init r=64) | $6.09 \pm 0.02$ | $67.28 \pm 0.29$ | $41.43 \pm 0.28$ |
|  | 64 (init r=128) | $6.16 \pm 0.02$ | $69.71 \pm 0.48$ | $43.82 \pm 0.39$ |
| LLM-Pruner | 8 (init r=64) | $6.09 \pm 0.03$ | $69.88 \pm 0.35$ | $42.25 \pm 0.32$ |
|  | 64 (init r=128) | $6.18 \pm 0.03$ | $70.88 \pm 0.45$ | $44.38 \pm 0.12$ |
| PrunedLoRA | 8 (init r=64) | $6.14 \pm 0.06$ | $69.02 \pm 0.53$ | $42.32 \pm 0.33$ |
|  | 64 (init r=128) | $6.19 \pm 0.04$ | $71.16 \pm 0.24$ | $44.32 \pm 0.11$ |
|  | 64 (init r=256) | $6.23 \pm 0.03$ | $72.21 \pm 0.45$ | $46.21 \pm 0.26$ |
|  | 64 (init r=512) | $\underline{6.25 \pm 0.06}$ | $\mathbf{74.88 \pm 0.42}$ | $\mathbf{48.31 \pm 0.24}$ |

Table 1: Performance comparison of fine-tuning and pruning baselines on MT-bench, GSM8K and HumanEval benchmarks for Llama-3-8B-Base Model. **Bold** indicates the best result, underline represents the second-best one. (↑: higher values indicate better performance)

Table 1 presents our experimental results, which highlight the effectiveness of pruning-based LoRA methods across different target ranks. With the small rank target, *PrunedLoRA* already retains much of the performance typically associated with higher-rank LoRA (r = 64). For example, *PrunedLoRA* at rank 8 achieves 69.02 on GSM8K and 42.32 on HumanEval, whose performance is close to LoRA with rank 64 (69.21 on GSM8K and 42.88 on HumanEval). It shows that pruning does not collapse model expressiveness even in the small rank regime.

As the rank increases, we observe a clear and consistent trend: all methods improve, and *PrunedLoRA* initialized from a larger over-parameterized rank (for instance 256 or 512) and pruned down to 64 yields substantial gains. Specifically, *PrunedLoRA* initialized at 256 and pruned to rank 64 reaches 72.21 on GSM8K and 46.21 on HumanEval, while initialization at 512 leads to 74.88 and 48.31. These results nearly close the gap to full fine-tuning, which attains 73.12 on GSM8K and 48.31 on HumanEval.

To keep a fair comparison, we also compare with other strong variants of LoRA in Appendix C.4. Our pruning-based strategy is not in conflict with these methods and can be naturally combined with them. In Appendix C.4, we provide additional experiments showing that integrating our approach

with LoRA variants leads to further performance gains, demonstrating that our technique enhances rather than replaces existing LoRA improvements.

| Method | Rank | Before (%) | After (%) | Memory | Training Time |
|---|---|---|---|---|---|
| Full FT | full rank | 100.00 | 100.00 | $\sim 8 \times 40G$ | 4h 23min |
| LoRA | 8 | 0.11 | 0.11 | $\sim 8 \times 17G$ | 2h 27min |
| LoRA | 64 | 0.84 | 0.84 | $\sim 8 \times 18G$ | 2h 28min |
| DoRA | 64 | 0.89 | 0.89 | $\sim 8 \times 18G$ | 2h 34min |
| AdaLoRA | 64 | 0.84 | 0.84 | $\sim 8 \times 19G$ | 2h 41min |
| PrunedLoRA (init r = 64) | 8 | 0.84 | 0.11 | $\sim 8 \times 19G$ | 2h 29min |
| PrunedLoRA (init r = 128) | 64 | 1.68 | 0.84 | $\sim 8 \times 20G$ | 2h 31min |
| PrunedLoRA (init r = 256) | 64 | 3.36 | 0.84 | $\sim 8 \times 22G$ | 2h 47min |
| PrunedLoRA (init r = 512) | 64 | 6.71 | 0.84 | $\sim 8 \times 28G$ | 3h 23min |

Table 2: Comparison of trainable parameter ratios, peak memory cost (per GPU on $8\times$H100 with FSDP), and training time across different fine-tuning methods.

**Memory and Time Costs.** In Table 2, we compare the percentage of trainable parameters (before and after pruning), peak memory cost and training time of our methods with full fine-tuning, LoRA, DoRA, and AdaLoRA on the math task and Llama-3-8B model. We measure memory cost using 8 H100 GPUs with batch size 1 following Wang et al. (2024b). As the step number of structured pruning is quite small in the overall fine-tuning step, we have a comparable training time. For example, for an initial rank of 512 in *PrunedLoRA*, the overall consuming time for pruning is 23 minutes. From Table 2, we make two key observations: (1) Even with a very high initial LoRA rank—such as 512—the peak memory consumption of *PrunedLoRA* remains substantially lower than that of full fine-tuning. (2) The additional computation introduced by the pruning procedure incurs only a mild overhead beyond the standard LoRA forward and backward passes. Empirically, LoRA with rank 64 requires 2h 28 min, while PrunedLoRA (target rank 8, initial rank 64) completes in 2h 29 min under the same setup. It demonstrates that the pruning step adds mild runtime cost.

**Results on Natural Language Understanding.** In Table 3, we report the GLUE benchmark results for different adaptation methods. Full fine-tuning remains the best baseline overall, achieving the best average score of 87.91. Our proposed *PrunedLoRA* method narrows the gap between low-rank adaptation and fine-tuning by increasing the initial rank.

| Method | MNLI ↑ | SST2 ↑ | CoLA ↑ | QNLI ↑ | MRPC ↑ | Average ↑ |
|---|---|---|---|---|---|---|
| Full FT | 86.33±0.06 | 94.75±0.21 | **80.70±0.24** | 93.19±0.22 | **84.56±0.73** | **87.91** |
| LoRA | 85.30±0.04 | 94.04±0.11 | 69.35±0.05 | 92.96±0.09 | 68.38±0.01 | 82.08 |
| DoRA | 85.67±0.09 | 94.04±0.53 | 72.04±0.94 | 93.04±0.06 | 68.08±0.51 | 82.57 |
| AdaLoRA | 85.45±0.11 | 93.69±0.20 | 69.16±0.24 | 91.66±0.05 | 68.14±0.28 | 81.62 |
| SparseGPT | 85.21±0.23 | 93.33±0.19 | 68.16±0.34 | 94.33±0.15 | 73.32±0.34 | 82.07 |
| LLM-Pruner | 84.76±0.12 | 93.12±0.30 | 65.21±0.25 | 93.39±0.33 | 76.43±0.31 | 82.18 |
| PrunedLoRA (init r = 128) | 85.21±0.32 | 93.21±0.29 | 73.43±0.23 | 93.34±0.12 | 74.21±0.18 | 83.48 |
| PrunedLoRA (init r = 256) | 86.21±0.09 | 94.21±0.31 | 74.43±0.32 | **94.55±0.05** | 78.21±0.28 | 85.12 |
| PrunedLoRA (init r = 512) | **86.67±0.12** | **95.22±0.34** | 78.43±0.45 | 93.45±0.25 | 84.19±0.34 | 87.19 |

Table 3: GLUE benchmark results with different adaptation methods. Best results are in **bold**, second-best are underlined. (↑: higher values indicate better performance).

## 4.2 EXPERIMENTS ON ABLATION STUDY

We conduct extensive ablation studies to better understand the design choices in *PrunedLoRA*. Detailed results are summarized in Appendix B.

**Initialization Rank and Scaling Factor.** We find that both the initialization rank and the scaling factor $\alpha$ critically affect the performance in Table 5. For a fixed rank, setting $\alpha$ proportional to the initialization rank yields the most stable convergence. For example, on GSM8K with rank 128, accuracy improves from 69.21 ($\alpha = r/2$) to 71.11 ($\alpha = r$), while larger values ($\alpha = 2r$) provide little additional gain. Increasing the initialization rank further enhances results, with the accuracy rising to

72.21 at $r = 512$ ($\alpha = r$). These results confirm the effectiveness of high-rank initialization combined with proportional scaling $\alpha$.

**Pruning Schedule.** We also vary the pruning interval ($K1$) and the number of columns pruned per step ($K2$) in Table 9. Gradual pruning with moderate intervals is consistently superior: pruning every 10 steps with $K2 = 2$ achieves the highest accuracy, while aggressive pruning ($K2 = 4$) slightly hurts performance. This suggests that maintaining stability during rank reduction is critical. Besides gradually pruning in post-training, we can also train LoRA with a high rank to converge and do *one-shot structure pruning* to obtain a low-rank adaptation (Appendix C.5).

**Target Rank.** Beyond the default rank budget 64 in LoRA, we also examine more aggressive compression (e.g., pruning to target rank in $\{8, 16\}$). As expected, extreme pruning leads to performance degradation, but *PrunedLoRA* remains competitive with or better than activation-based and simple gradient-based baselines at the same target rank (see Appendix C.3). This highlights the robustness of structured pruning with the awareness of the overall under the cases of extreme compression.

**Comparison with High-Rank LoRA.** Since *PrunedLoRA* compresses high-rank LoRA modules into a compact low-rank representation, it is essential to examine how much performance degradation this compression introduces. As shown in Table 4, pure high-rank LoRA exhibits a clear upward trend from rank 128 to 512 on both GSM8K and HumanEval, reflecting increased representational capacity with larger ranks. Remarkably, *PrunedLoRA* initialized at a sufficiently high rank (e.g., $r=512$) matches the performance of LoRA-512 almost exactly (74.88 vs. 74.98 on GSM8K and 48.31 vs. 48.49 on HumanEval), demonstrating that second-order pruning preserves most of the expressive power of the original adapter. Even with a moderate initialization (e.g., $r=256$), PrunedLoRA retains the vast majority of the performance of LoRA with rank 256, with only minor degradation (less than 0.3 HumanEval). These results collectively indicate that *PrunedLoRA* effectively distills high-rank LoRA into a compact low-rank form with minimal performance loss.

| Method | Target Rank | GSM8K ↑ | HumanEval ↑ |
|---|---|---|---|
| LoRA | 128 | 72.51 | 44.82 |
| LoRA | 256 | 73.83 | 46.48 |
| LoRA | 512 | 74.98 | 48.49 |
| PrunedLoRA (init $r = 128$) | 64 | 71.16 | 44.32 |
| PrunedLoRA (init $r = 256$) | 64 | 72.21 | 46.21 |
| PrunedLoRA (init $r = 512$) | 64 | 74.88 | 48.31 |

Table 4: Comparison between LoRA and PrunedLoRA at different target and initial ranks.

## 5 CONCLUSION

In this work, we introduced *PrunedLoRA*, a gradient-based structured pruning framework for obtaining efficient low-rank adapters from over-parameterized spaces. By formulating pruning as an optimization problem that explicitly minimizes the loss induced by weight perturbations, our method provides a theoretically grounded strategy for structured adapter compression. Comprehensive experiments on mathematical reasoning, code generation, and natural language understanding demonstrate that *PrunedLoRA* consistently narrows the gap to full fine-tuning while retaining inference efficiency. Furthermore, across diverse sparsity levels, it achieves superior performance over existing structured pruning baselines, underscoring both its robustness and practical effectiveness.

## ETHICS STATEMENT

This work does not involve human subjects, sensitive data, or any practices that could raise ethical concerns. We confirm that our study complies with the ICLR Code of Ethics and does not present any potential violations.

## REPRODUCIBILITY STATEMENT

To ensure reproducibility, we provide implementation details in Sec. 4.1. The code is available at the following anonymized link `https://anonymous.4open.science/r/PrunedLoRA-FED0/README.md`.

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

## A  THE USE OF LLMs

LLMs were used to improve writing clarity and assist with code development. Specifically, LLMs assisted in improving the clarity, fluency, and grammatical correctness of the manuscript, including rephrasing sentences and ensuring consistent terminology. Additionally, LLMs helped generate auxiliary code and scripts for data processing, experimental setup, and result visualization. However, the core research ideas, technical contributions, experimental design, and scientific conclusions are entirely the intellectual contribution of the human authors. All LLM-generated content underwent thorough human review and verification to ensure technical accuracy, scientific rigor, and alignment with our research objectives.

## B  ANALYSIS FOR STRUCTURED PRUNING STRATEGIES

In this section, we provide supplementary details and additional analysis complementing Sec. 3. Appendix B.1 presents the formal statement of Proposition 1 together with its proof, which underscores the robustness of gradient-based structural pruning methods with respect to the overall loss. Furthermore, Appendix B.3 analyzes the minimizer of Problem (6) and describes the procedure for pruning columns of a full weight matrix, as summarized in Algorithm 2.

### B.1  ANALYSIS FOR GRADIENT-BASED PRUNING VERSUS ACTIVATION-BASED PRUNING

As discussed in Sec. 2, structured pruning strategies can be broadly categorized into two classes, both of which are widely adopted in foundation model compression (Hubara et al., 2021; Kurtic et al., 2025; Wu et al., 2024; Frantar & Alistarh, 2022). To better understand their implications, we provide a theoretical analysis examining how these strategies affect the overall loss. Since different approaches employ distinct criteria to measure precision, we first formalize the notion of perturbation error and analyze its influence on predictive performance. Let $W \in \mathbb{R}^{m \times n}$ denote the original weight matrix and $\widehat{W}$ its pruned counterpart. While our discussion primarily focuses on structured pruning, we note that our analysis, in principle, can be extended to non-structured settings.

It is important to highlight a key distinction between the two classes of methods for the sake of conceptual clarity. Although activation-based approaches can also apply a Taylor expansion and obtain the first-order gradient term, this gradient arises from the reconstruction objective rather than from the overall loss. In contrast, gradient-based pruning methods explicitly leverage the gradient of the overall loss, providing a more direct connection to the model's predictive performance.

**Definition 1** ($\varepsilon$-**Perturbation Error**) *We define the perturbation error under different pruning criteria as follows:*

- *For **activation-based** pruning strategies, we say the pruned weight matrix $\widehat{W}$ satisfies $\varepsilon$-perturbation error if: $\|\widehat{W}X - WX\| \leq \varepsilon$, where $X$ is the input of the parameter layer.*

- *For **gradient-based** pruning strategies, we define $\varepsilon$-perturbation error as: $|\mathcal{L}(\widehat{W}) - \mathcal{L}(W)| \leq \varepsilon$, where $\mathcal{L}$ denotes the task-specific loss function.*

In Def 1, the metrics of perturbation error for activation-based pruning and gradient-based pruning strategies derive from (1) and (2), respectively. Noticeably, even though we can set the same precision of the perturbation error for different pruning strategies (under Def 1 ), we cannot know how the perturbation error of different pruning strategies contributes to the overall loss. Intuitively, gradient-based strategies emphasize preserving the global correlation between $\widehat{W}$ and $W$, which suggests greater robustness to weight perturbations for the overall loss. However, this intuition has not yet been formally established. In the following, we conduct an analysis on a single attention module to provide theoretical justification for this claim. It is an official statement of Proposition 1.

**Proposition 2** *(Official Statement) In a single attention module, if we assume each module of $(Q, K, V)$ satisfying perturbation error $\varepsilon$ in activation-based strategies, respectively, the overall loss would be linear w.r.t the perturbation error up to the magnitude of each module. However, if they satisfy the perturbation error $\varepsilon$ in gradient-based strategies, the overall loss would be linear the perturbation error and independent of the magnitude for each module.*

*Proof:* Given an input $X \in \mathbb{R}^{n \times d_{\text{model}}}$, the query, key, and value module of a single attention module are obtained through three separate linear transformations:

$$Q = XW_Q, \quad K = XW_K, \quad V = XW_V,$$

where $W_Q, W_K, W_V \in \mathbb{R}^{d_{\text{model}} \times d}$ are trainable weight matrices, and $d$ is the dimensionality of a single attention head. Here, we assume these three modules have the same dimension. The attention output is then computed as

$$Z = \text{softmax}\left(\frac{QK^\top}{\sqrt{d}}\right) V.$$

The scaling factor $1/\sqrt{d}$ is introduced to prevent $QK^\top$ from growing too large in magnitude, which would otherwise make the softmax distribution extremely peaked and lead to unstable gradients. Given a weight vector $(x_1, x_2, \cdots, x_d)$, the softmax function will transform the $i$-th element in the vector as

$$\text{softmax}(x_i) = \frac{\exp(x_i)}{\sum_j \exp(x_j)},$$

which transforms a vector of real numbers into a probability distribution. In the attention mechanism, the softmax ensures that the attention weights assigned to all keys are non-negative and sum to one.

First, we will analyze activation-based pruning strategies. If we suppose $\|Q - \widehat{Q}\|_F \leq \varepsilon, \|K - \widehat{K}\|_F \leq \varepsilon, \|V - \widehat{V}\|_F \leq \varepsilon$, respectively, i.e., perturbation error in each module is bounded by $\varepsilon$ (See Def 1). Then,

$$\left\|Z - \widehat{Z}\right\|_F \leq \left\|A(V - \widehat{V})\right\|_F + \left\|(A - \widehat{A})\widehat{V}\right\|_F,$$

where $A = \text{softmax}\left(\frac{QK^\top}{\sqrt{d}}\right)$ and $\widehat{A} = \text{softmax}\left(\frac{\widehat{Q}\widehat{K}^\top}{\sqrt{d}}\right)$. The first term is at most $\varepsilon$ due to the fact that $\|A\| \leq 1$. The second term depends on the mismatch between $Q$ and $K$ after pruning:

$$\left\|QK^\top - \widehat{Q}\widehat{K}^\top\right\|_F \leq \|Q\| \cdot \left\|K - \widehat{K}\right\|_F + \|K\| \cdot \left\|Q - \widehat{Q}\right\|_F.$$

This shows that the error in $A$ scales linearly with both $\varepsilon$ and the magnitude of $Q$ and $K$, leading to an overall bound:

$$\left\|Z - \widehat{Z}\right\|_F \leq \left(1 + \frac{\|Q\| + \|K\|}{\sqrt{d}} \cdot \|\widehat{V}\|\right) \varepsilon$$

In contrast, under the perturbation error of gradient-based tuning strategies, if we assume that $\mathcal{L}(Q, K, V)$ is the loss of a single attention module, we know that

$$\left|\mathcal{L}(Q, K, V) - \mathcal{L}(\widehat{Q}, \widehat{K}, \widehat{V})\right| \leq 3\varepsilon,$$

which is a direct consequence of the triangle inequality. This concludes the proof.

Next, we will analyze how pruning a single weight matrix $\boldsymbol{W}$ affects the overall loss function $\mathcal{L}$ in the general cases. Assume that the loss function $\mathcal{L}$ is $C$-Lipschitz continuous (see (Federer, 2014; Latorre et al., 2020) for formal definitions).

For gradient-based pruning methods, if the pruning procedure introduces an $\varepsilon$-level perturbation error to the weights, the resulting loss change is at most $\varepsilon$, i.e., the approximation error in the loss is directly proportional to the perturbation error. This result is consistent with the conclusion we established on the toy model.

In contrast, for activation-based pruning methods, pruning a weight matrix with perturbation error $\varepsilon$ yields a change in the loss that is bounded by $C\varepsilon$, where $C$ is the Lipschitz constant of $\mathcal{L}$. Recent work (Khromov & Singh, 2023) has shown that both the lower and upper bounds of the Lipschitz constant tend to increase as training progresses. Consequently, the sensitivity of the loss to perturbations induced by activation-based pruning can escalate over the course of fine-tuning, making its impact more difficult to control compared to gradient-based approaches.

Therefore, in the toy model, we can explicitly observe the impact of pruning multiple matrices under both gradient-based and activation-based strategies. The larger the matrix magnitude, the greater the error inflation in the overall loss function in activation-based methods. More generally, when considering a single weight matrix in any loss function, our analysis also highlights that activation-based methods are influenced by the Lipschitz constant, in contrast to gradient-based methods.

## B.2 EXPERIMENT FOR PROPOSITION 2

To empirically verify the robustness difference stated in Proposition 2, we conduct a controlled synthetic experiment using a self-attention model trained on a linear regression task. The goal is to examine how *activation-based* and *gradient-based* pruning strategies impact the change of loss under the same perturbation constraint $\varepsilon$.

**Data generation.** We draw covariates $X \in \mathbb{R}^{n \times d}$ (rows are samples) i.i.d. from a zero-mean sub-Gaussian distribution (standard normal in our implementation), and generate responses

$$y = Xw^\star + \xi \in \mathbb{R}^{n \times 1}, \qquad \xi \sim \mathcal{N}(0, \sigma^2 I_n),$$

with $(n, d) = (2000, 32)$. We use mean squared error (MSE) as the task loss:

$$\mathcal{L}(\Theta) = \frac{1}{n} \left\| f_\Theta(X) - y \right\|_2^2.$$

**Model architecture.** We use a single layer of self-attention with a linear layer for prediction for modeling. Given an input matrix $X \in \mathbb{R}^{n \times d}$ with $n = 2000$ and $d = 32$, the model predicts scalar responses $\widehat{y} \in \mathbb{R}^{n \times 1}$ through three stages:

1. **Input projection.** The raw features are first linearly projected to obtain $H \in \mathbb{R}^{n \times d}$:

$$H = XW_{\mathrm{in}} + \mathbf{1}b_{\mathrm{in}}^\top,$$

where $W_{\mathrm{in}} \in \mathbb{R}^{d \times d}$ and $b_{\mathrm{in}} \in \mathbb{R}^d$.

2. **Single-head self-attention.** The attention block contains three trainable weight matrices:

$$W_Q, W_K, W_V \in \mathbb{R}^{d \times d}.$$

Queries, keys, and values are computed as

$$Q = HW_Q, \qquad K = HW_K, \qquad V = HW_V.$$

Attention weights are obtained via scaled dot-product:

$$A = \mathrm{softmax}\left( \frac{QK^\top}{\sqrt{d}} \right) \in \mathbb{R}^{n \times n},$$

and the attention output is the weighted aggregation of values:

$$Z = AV \in \mathbb{R}^{n \times d}.$$

3. **Output projection.** The output head is a single linear transformation

$$\widehat{y} = ZW_{\mathrm{out}} + \mathbf{1}b_{\mathrm{out}} \in \mathbb{R}^{n \times 1},$$

with $W_{\mathrm{out}} \in \mathbb{R}^{d \times 1}$ and $b_{\mathrm{out}} \in \mathbb{R}$.

The full parameter set is therefore

$$\Theta = \{W_{\mathrm{in}}, b_{\mathrm{in}}, W_Q, W_K, W_V, W_{\mathrm{out}}, b_{\mathrm{out}}\},$$

which includes five linear weight matrices and two bias vectors. Then the model is trained using the mean-squared-error objective

$$\mathcal{L}(\Theta) = \frac{1}{n} \left\| \widehat{y} - y \right\|_2^2, \tag{12}$$

optimized with Adam ($\mathrm{lr} = 10^{-3}$) for 2000 iterations until convergence. The learned attention weights $\{W_Q, W_K, W_V, W_{\mathrm{out}}\}$ are later used as the pruning targets in our robustness comparison experiments.

**Activation-based pruning (SparseGPT-style).** For each projection matrix $W$ with corresponding input activations $A_{\text{in}}$, we seek a sparse approximation $\widehat{W}$ that preserves the layer output within a small reconstruction error tolerance. Formally, we minimize the activation reconstruction discrepancy

$$\text{L}_{\text{act}}(A_{\text{in}}, W, \widehat{W}) = \big\| A_{\text{in}}(W - \widehat{W}) \big\|_F \quad \text{s.t.} \quad \text{L}_{\text{act}}(A_{\text{in}}, W, \widehat{W}) \leq \varepsilon,$$

where $\varepsilon$ controls the acceptable deviation in the forward activations. Practically, we solve a sequence of column-wise least-squares subproblems on $A_{\text{in}}$, greedily selecting the most significant columns of $W$ (in analogy to SparseGPT). We stop once the prescribed tolerance is reached.

**Gradient-based pruning (LLM-Pruner-style).** In contrast, gradient-based pruning leverages the first-order sensitivity of the loss function with respect to the model parameters. For each weight matrix $W \in \mathcal{W}$, we compute its gradient $G = \partial \mathcal{L} / \partial W$ and assign elementwise Taylor saliency scores

$$s_{ij} = |W_{ij} G_{ij}|.$$

Parameters with the smallest saliency scores are progressively pruned until the total parameter perturbation satisfies

$$\text{L}_{\text{w}}(W, \widehat{W}) = \big\| \mathcal{L}(W) - \mathcal{L}(\widehat{W}) \big\|_F \leq \varepsilon.$$

This approach explicitly bounds the norm of the weight perturbation rather than the activation mismatch, ensuring that the induced change in loss remains linearly proportional to $\varepsilon$.

**Results.** Solving the prediction objective in (12) yields an estimated model with a baseline loss of $6.52 \times 10^{-4}$. We then apply both pruning strategies under matched precision constraints.

For the *gradient-based* method, enforcing the same perturbation budget produces pruned projection matrices that increase the loss only slightly, to $8.25 \times 10^{-4}$. This corresponds to a negligible degradation, indicating that directly constraining the weight perturbation effectively preserves the model's predictive behavior.

In contrast, the *activation-based* strategy yields a substantially larger post-pruning loss of $2.23 \times 10^{-3}$, despite operating under an equivalent tolerance. This degradation—over three times larger than the gradient-based counterpart—highlights the instability of activation reconstruction as a pruning criterion: activation mismatch can propagate and be amplified through the network, making it markedly less robust under the same nominal precision level.

### B.3 ANALYSIS FOR THE MASKING PRUNING AND WEIGHT UPDATE IN THE PROBLEM 6

In this part, we will provide a detailed analysis of the Problem (6) as

$$\mathcal{M}_s, \boldsymbol{\delta} = \arg\min_{\mathcal{M}_s, \boldsymbol{\delta}} \langle \nabla_{\boldsymbol{W}} \mathcal{L}, \boldsymbol{\delta} \rangle + \frac{1}{2} tr(\boldsymbol{\delta} \widehat{\boldsymbol{H}} \boldsymbol{\delta}^{\top}) \tag{13}$$
$$s.t. \quad \boldsymbol{\delta}_{:, \mathcal{M}_s} = -\boldsymbol{W}_{:, \mathcal{M}_s}.$$

with optimal solutions for pruning selection $\mathcal{M}_s$ and weight update $\boldsymbol{\delta}$.

Here, for simplicity, we denote $\nabla_{\boldsymbol{W}} \mathcal{L}(\boldsymbol{W})$ as $\nabla_{\boldsymbol{W}} \mathcal{L}$. The corresponding Lagrange problem is

$$\langle \nabla_{\boldsymbol{W}} \mathcal{L}, \boldsymbol{\delta} \rangle + \frac{1}{2} tr(\boldsymbol{\delta} \widehat{\boldsymbol{H}} \boldsymbol{\delta}^{\top}) + \langle \boldsymbol{\Lambda}, (\boldsymbol{\delta})_{:, \mathcal{M}_s} + \boldsymbol{W}_{:, \mathcal{M}_s} \rangle, \tag{14}$$

where $\boldsymbol{\Lambda} \in \mathbb{R}^{m \times n}$ is a Lagrange multiplier. Under first order condition of $\boldsymbol{\delta}$, it implies

$$\nabla_{\boldsymbol{W}} \mathcal{L} + \boldsymbol{\delta} \widehat{\boldsymbol{H}} + \boldsymbol{\Lambda} P_{\mathcal{M}_s} = 0, \tag{15}$$

where $P_{\mathcal{M}_s} \in \mathbb{R}^{n \times n}$ is a diagonal matrix whose $i$-th diagonal entry is 1 if the $i$-th column is pruned and 0 otherwise. Then we have

$$\boldsymbol{\delta} = -\left(\nabla_{\boldsymbol{W}} \mathcal{L} + \boldsymbol{\Lambda} P_{\mathcal{M}_s}\right) \widehat{\boldsymbol{H}}^{-1} = -\nabla_{\boldsymbol{W}} \mathcal{L} \, \widehat{\boldsymbol{H}}^{-1} - \boldsymbol{\Lambda} P_{\mathcal{M}_s} \, \widehat{\boldsymbol{H}}^{-1}. \tag{16}$$

Then we could put the expression of $\boldsymbol{\delta}$ back into the structure constraint (3) and get

$$\boldsymbol{\Lambda} = \left(\boldsymbol{W}_{:, \mathcal{M}_s} - (\nabla_{\boldsymbol{W}} \mathcal{L} \, \widehat{\boldsymbol{H}}^{-1})_{:, \mathcal{M}_s}\right) \left((\widehat{\boldsymbol{H}}^{-1})_{\mathcal{M}_s, \mathcal{M}_s}\right)^{-1}. \tag{17}$$

---

**Algorithm 2** Gradient-based structured pruning with Weight Update. We prune the layer matrix $\boldsymbol{W}$ with column-wise sparsity $s$ given the gradient $\nabla_{\boldsymbol{W}}\mathcal{L}$ and the Hessian matrix $\widehat{\boldsymbol{H}} = (\nabla_{\boldsymbol{W}}\mathcal{L})^T \nabla_{\boldsymbol{W}}\mathcal{L}$

---

1: **Step 1: Search pruning columns with sparsity $s$.**

$$\arg\min_{\mathcal{M}_s}\ \mathrm{tr}\left((\boldsymbol{W} - \nabla_{\boldsymbol{W}}\mathcal{L}\,\widehat{\boldsymbol{H}}^{-1})_{:,\mathcal{M}_s}\left((\widehat{\boldsymbol{H}}^{-1})_{\mathcal{M}_s,\mathcal{M}_s}\right)^{-1}(\boldsymbol{W} - \nabla_{\boldsymbol{W}}\mathcal{L}\,\widehat{\boldsymbol{H}}^{-1})_{:,\mathcal{M}_s}^{\top}\right).$$

2: **Step 2: Compute optimal update.**
3: Given $\mathcal{M}_s$, compute update $\boldsymbol{\delta}$:

$$\boldsymbol{\delta} = -\nabla_{\boldsymbol{W}}\mathcal{L}\widehat{\boldsymbol{H}}^{-1} - (\boldsymbol{W} - \nabla_{\boldsymbol{W}}\mathcal{L}\widehat{\boldsymbol{H}}^{-1})_{:,\mathcal{M}_s}\left((\widehat{\boldsymbol{H}}^{-1})_{\mathcal{M}_s,\mathcal{M}_s}\right)^{-1}(\widehat{\boldsymbol{H}}^{-1})_{\mathcal{M}_s,:}$$

4: **Step 3: Update model.**
5: Set $\boldsymbol{W} \leftarrow \boldsymbol{W} + \boldsymbol{\delta}$.
6: **Step 4: Iterate or finalize.**
7: If multi-round pruning, repeat Steps 1–3 until target sparsity/rank is reached. Otherwise, output $\boldsymbol{W}$.

---

Finally, putting the form of $\boldsymbol{\Lambda}$ in (17) back into (15), we could get $\boldsymbol{\delta}$ as

$$\begin{aligned}
\boldsymbol{\delta} = &-\nabla_{\boldsymbol{W}}\mathcal{L}\,\widehat{\boldsymbol{H}}^{-1} - \boldsymbol{W}_{:,\mathcal{M}_s}\left((\widehat{\boldsymbol{H}}^{-1})_{\mathcal{M}_s,\mathcal{M}_s}\right)^{-1}(\widehat{\boldsymbol{H}}^{-1})_{\mathcal{M}_s,:} \\
&+ (\nabla_{\boldsymbol{W}}\mathcal{L}\widehat{\boldsymbol{H}}^{-1})_{:,\mathcal{M}_s}\left((\widehat{\boldsymbol{H}}^{-1})_{\mathcal{M}_s,\mathcal{M}_s}\right)^{-1}(\widehat{\boldsymbol{H}}^{-1})_{\mathcal{M}_s,:}.
\end{aligned} \tag{18}$$

structured pruning methods (Liu et al., 2017; Nova et al., 2023; Yang et al., 2025) remove entire structured components of a network, facilitating efficient GPU speedups Li et al. (2025b). Utilizing the gradient of the overall loss function in training, termed gradient-based methods, can be robust for eliminating the change of loss under the impact of weight perturbation in pruning. Gradients of weight are computed during the normal optimization process; one can easily reuse those for determining weight importance efficiently. Within the context of gradient-based pruning, we want to further explain the development of existing methods and clarify the difference with our effort in this paper. Most of the works in the literature use an important score to select the pruning structure (Molchanov et al., 2019b; Zhang et al., 2023; Shen et al., 2022; Fang et al., 2023; Molchanov et al., 2019a). They provide refined pruning selection but do not further eliminate the influence of structured pruning. (Xia et al., 2022) combines distillation with pruning to improve performance and erase the impact of structured pruning, but they require minimizing the KL-divergence of two distributions and cannot find a closed-form solution.

Inspired by Optimal Brain Surgeon, (Singh & Alistarh, 2020; Kurtic et al., 2022; Das et al., 2023) propose a weight update after model pruning in the context of model compression to further eliminate the influence of pruning. Since their analysis is established for one-dimensional weight vectors, the pruning metric is hard to interpret. In contrast, we establish the analysis for the weight matrix and provide a grounded interpretation for the pruning selection and weight update (See Sec 3.2).

## C EXPERIMENT

### C.1 THE DETAILS ABOUT NATURAL LANGUAGE GENERATION TASK

**Dialogue Generation Task** We fine-tune large language models on a 52k subset of the WizardLM dataset Xu et al. (2024) and evaluate it using the MT-Bench dataset Zheng et al. (2023). GPT-4o is used to assess the quality of the model's response and we report the first-turn score as the metric.

**Math Task** We fine-tune large language models on a 100k sample from the MetaMathQA dataset Yu et al. (2023). The model is then evaluated on the GSM8K test set Cobbe et al. (2021), and we report the accuracy as the metric.

**Coding Task** We fine-tune large language models on a 100k subset of the CodeFeedback dataset Zheng et al. (2024) and test it on the HumanEval dataset Chen et al. (2021), reporting the PASS@1 metric.

We fine-tune each task for three epochs, with a maximum of 5000 training steps. All experiments are conducted on NVIDIA H100 GPU cards.

## C.2 PRUNING STRATEGY

**Dynamic Pruning.** Motivated by Fig. 1, we observe that higher-rank LoRA adapters ($A$ and $B$) achieve better empirical performance with smaller variance. Based on this observation, we propose to prune adapters starting from higher-rank spaces. Specifically, we initialize adapters with rank $r \in \{128, 256, 512\}$ and progressively prune them down to rank 64, corresponding to 50%, 75%, and 87.5% sparsity, respectively. We also explore more aggressive settings (e.g., pruning from $r$ to 8). Pruning is performed in a structured manner, controlled by two hyperparameters: the pruning interval $k_1$ and the number of columns removed per step $k_2$. For example, with $k_1 = 10$ and $k_2 = 2$, we prune two columns every ten training steps. Once the remaining columns reach the target rank budget (default: 64), pruning is terminated.

**Adaptive Choice of Hyperparameter.** Importantly, as rank dynamically changes during training, the scaling factor $\alpha$ must remain stable. While vanilla LoRA typically sets $\alpha = 16$, we find this choice suboptimal for higher-rank initializations. To address it, we perform a grid search over a large range and identify that $\alpha \in \{r/2, r, 2r\}$ can achieve the better performance, where $r$ is the current rank in LoRA. The hyperparameter $\alpha$ will be proportional to $r$ over the training process.

## C.3 ABLATION STUDY

| Init Rank | $\alpha$ | GSM8K Acc. | Loss |
|-----------|----------|------------|------|
| 128 | 64 | 69.21 | 0.13 |
| 128 | 128 | 71.11 | 0.14 |
| 128 | 256 | 71.16 | 0.14 |
| 256 | 128 | 71.88 | 0.12 |
| 256 | 256 | 72.21 | 0.11 |
| 256 | 512 | 71.01 | 0.13 |
| 512 | 256 | 74.21 | 0.11 |
| 512 | 512 | 74.21 | 0.10 |
| 512 | 1024 | 73.99 | 0.12 |

Table 5: Ablation study of *PrunedLoRA* on GSM8K with different initial ranks and scaling factors $\alpha$ (rank/2, rank, 2×rank). Each row reports Accuracy and the final training loss.

**Hyperparamter $\alpha$ and Initial Rank.** To better understand the sensitivity of *PrunedLoRA* to the initial rank and the scaling factor $\alpha$, we conduct an ablation study on GSM8K with different settings of Init r $\in \{128, 256, 512\}$ and scaling factor $\alpha \in \{r/2, r, 2r\}$, where $r$ denotes the current rank. Table 5 reports the results, with each row showing accuracy and loss. It shows that both the initialization rank and the scaling factor $\alpha$ play a critical role in the performance of *PrunedLoRA*. For a fixed rank, setting $\alpha = r$ yields the best trade-off between accuracy and stability, while smaller values under-scale the updates and larger values bring little additional gain. Moreover, larger initialization ranks consistently improve results, with accuracy increasing from 72.21 at $r = 128$ to 74.21 at $r = 512$ when $\alpha = r$. These findings confirm that *PrunedLoRA* benefits from high-rank initialization and that scaling $\alpha$ proportionally to the rank is the most effective choice.

**Comparison of Pruning Strategies under Different Initialization Ranks.** Table 6 reports the performance of SparseGPT, LLM-Pruner, and PrunedLoRA with different initialization ranks ($r = 128, 256, 512$). We observe that while all methods benefit from larger initial ranks, the gains are much more pronounced for *PrunedLoRA*, which achieves the best performance at $r = 512$. It further supports the effectiveness of gradient-based pruning over other structured pruning methods.

| Method | Init r | GSM8K | HumanEval |
|--------|--------|-------|-----------|
| *SparseGPT* | 128 | 69.71±0.48 | 43.82±0.39 |
| | 256 | 69.88±0.34 | 44.12±0.10 |
| | 512 | 72.12±0.48 | 43.12±0.36 |
| *LLM-Pruner* | 128 | 70.88±0.45 | 44.38±0.12 |
| | 256 | 71.21±0.17 | 44.67±0.29 |
| | 512 | 74.19±0.23 | 46.21±0.23 |
| *PrunedLoRA* | 128 | 71.16±0.24 | 44.32±0.11 |
| | 256 | 72.21±0.45 | 44.32±0.11 |
| | 512 | **74.88±0.42** | **48.31±0.24** |

Table 6: Comparison of *SparseGPT*, *LLM-Pruner*, and *PrunedLoRA* under different initial ranks on GSM8K and HumanEval benchmarks using Llama-3-8B-Base. **Bold** indicates the best result, underline represents the second-best one.

**Pruning Schedule $K_1$ and $K_2$.** We further investigate the impact of the pruning schedule on the performance of *PrunedLoRA*. Specifically, we vary the pruning interval $K_1 \in \{5, 10\}$, which controls how frequently pruning is applied, and the number of columns pruned at each step $K_2 \in \{2, 4\}$. Table 9 summarizes the results on GSM8K. We find that less frequent pruning with a smaller number of pruning indices at each pruning step (e.g., $K_1 = 10$, $K_2 = 2$) leads to stable performance, while larger $K_2$ values slightly hurt accuracy. It suggests that gradual pruning with moderate intervals achieves better performance.

| $K_1$ | $K_2$ | GSM8K |
|-------|-------|-------|
| 5 | 2 | 69.39 |
| 5 | 4 | 70.23 |
| 10 | 2 | 71.16 |
| 10 | 4 | 71.12 |

Table 7: *PrunedLoRA* on GSM8K with different pruning schedules. $K_1$ is the pruning interval (steps between pruning), and $K_2$ is the number of pruning indices at each step.

**Pruning for Different Low-rank Targets.** We further investigate the effect of initialization rank and pruning budget on downstream performance. Figures 3 presents results where LoRA adapters are initialized with $r = 512, 256, 128, 64$ and pruned to smaller target budgets ($r = 32, 16, 8$). Across all settings, *PrunedLoRA* consistently outperforms classical one-shot pruning approaches such as SparseGPT and LLM-Pruner, and maintains accuracy close to or above the unpruned LoRA baseline. The performance gap becomes more pronounced when the pruning ratio is high (e.g., pruning lora from the init r 128 to the target rank 8), highlighting that gradient-informed structured pruning is more robust under extreme compression. These results confirm that *PrunedLoRA* provides both stability and generalization, making it preferable when adapting to stringent memory and efficiency constraints.

## C.4    OTHER VARIANTS OF LORA

### C.4.1    OPTIMIZATION IN LORA

In this part, we focus on comparing *PrunedLoRA* with several representative LoRA variants listed below, and *we emphasize that our method does not conflict with these approaches*. Instead, it can be naturally combined with them to further narrow the gap between low-rank adapters and full fine-tuning.

- **LoRA+** (Hayou et al., 2024): setting different learning rates for the LoRA adapter matrices $A$ and $B$ with a well-chosen fixed ratio.
- **LoRA-GA** (Wang et al., 2024a): make the gradient of first step in LoRA align with the full fine-tuning by modifying the initlization of LoRA.

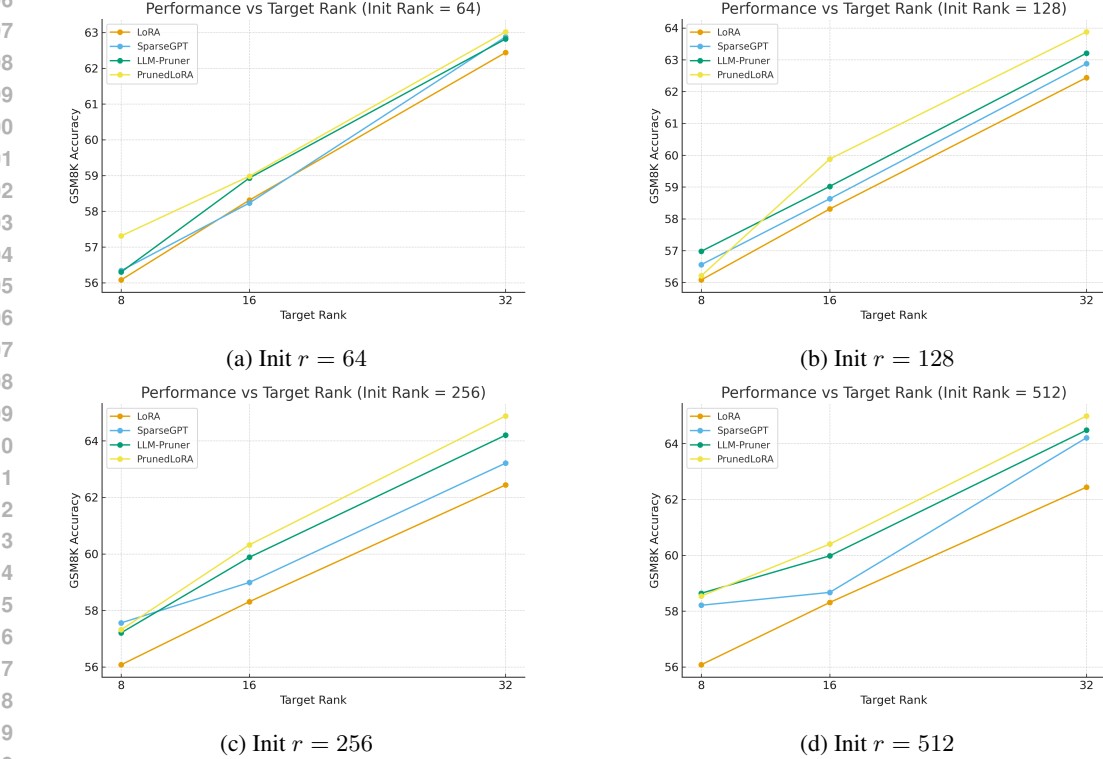

Figure 3: GSM8K accuracy of different pruning methods (SparseGPT, LLM-Pruner, and Pruned-LoRA) under various initialization ranks $r \in 64, 128, 256, 512$ and target ranks $8, 16, 32$. Each subfigure reports performance when starting from a specific initialization rank.

- **LoRA-Pro** (Wang et al., 2024b): project the full gradient to the low-rank space of $A$ and $B$ jointly.
- **AltLoRA** (Yu et al., 2025): alternatingly project the full gradient to the low-rank spaces of $A$ and $B$ with grounded theoretical guarantee.

| $Methods$ | Target Rank | **GSM8K** |
|---|---|---|
| LoRA+ | 8 | $71.48 \pm 1.23$ |
| LoRA-GA | 8 | $71.38 \pm 0.83$ |
| LoRA-Pro | 8 | $71.12 \pm 0.23$ |
| AltLoRA | 8 | $73.32 \pm 0.31$ |
| LoRA+ | 64 | $72.89 \pm 1.11$ |
| LoRA-GA | 64 | $73.01 \pm 0.92$ |
| LoRA-Pro | 64 | $71.73 \pm 0.32$ |
| AltLoRA | 64 | $73.88 \pm 0.18$ |
| PrunedLoRA + LoRA+ (init r 64) | 8 | $72.29 \pm 1.01$ |
| PrunedLoRA + LoRA-GA (init r 64) | 8 | $72.33 \pm 0.79$ |
| PrunedLoRA + LoRA-Pro (init r 64) | 8 | $71.94 \pm 0.34$ |
| PrunedLoRA + AltLoRA (init r 64) | 8 | $73.45 \pm 0.28$ |

Table 8: Performance of LoRA variants at ranks 8 and 64, and their combinations with *PrunedLoRA*. Initializing at rank 64 and pruning to rank 8 consistently recovers part of the high-rank gain and improves all baseline variants.

Even though these LoRA variants already achieve strong performance at a small target rank of 8, their performance gains do not saturate as the rank increases—higher ranks consistently offer additional benefits. By integrating our *PrunedLoRA* technique, we are able to recover part of this "higher-rank gain" even when the final target rank remains 8. As shown in Table 8, combining

PrunedLoRA with LoRA+, LoRA-GA, LoRA-Pro, and AltLoRA consistently improves their rank-8 results, demonstrating that our pruning strategy is complementary to these optimization-oriented LoRA variants and can further enhance their effectiveness in the low-rank regime.

### C.4.2 ARCHITECTURE IN LORA

Besides LoRA-optimization–oriented methods, there also exist strong baselines that explicitly aim to increase the expressiveness or effective rank of PEFT adapters, including HiRA (Huang et al., 2025), MoRA (Jiang et al., 2024), and ABBA (Singhal et al., 2025), which have benn briefly summarized below.

- **HiRA**: enhances the expressiveness of low-rank adapters by applying a Hadamard product to reconstruct a higher-rank update. However, its effectiveness heavily depends on the structural properties of the pre-trained weight matrix.
- **MoRA**: introduces a square matrix to enable higher-rank updates while keeping the number of trainable parameters unchanged. Yet, the final update matrix contains many zero entries, which limits its ability to adapt LLMs across diverse downstream tasks.
- **ABBA**: models the update as a Hadamard product between two independently learned low-rank matrices. Similar to HiRA, ABBA focuses only on the *upper bound* of the achievable rank, while providing no *lower bound* guarantees.

In contrast to our methods, all these methods shall modify the architecture of low-rank adapters for implementation, which requires additional effort in the common work pipeline like huggingface..

| Methods | Target Rank | GSM8K |
|---------|-------------|-------|
| LoRA | 8 | $65.27 \pm 0.13$ |
|  | 64 | $69.21 \pm 0.36$ |
| HiRA | 8 | $67.38 \pm 0.09$ |
|  | 64 | $72.01 \pm 0.19$ |
| MoRA | 8 | $68.88 \pm 0.11$ |
|  | 64 | $71.08 \pm 0.21$ |
| ABBA | 8 | $67.78 \pm 0.18$ |
|  | 64 | $72.23 \pm 0.24$ |
| PrunedLoRA | 8 | $69.02 \pm 0.12$ |
|  | 64 | $71.16 \pm 0.34$ |
|  | 64 (init r = 256) | $72.21 \pm 0.45$ |
|  | 64 (init r = 512) | $74.88 \pm 0.42$ |

Table 9: Comparison between *PrunedLoRA* and expressiveness-enhancing PEFT variants (HiRA, MoRA, ABBA) on GSM8K with target ranks 8 and 64 using the pretrained Llama-3-8B base model.

Note that under the target rank of 8, expressiveness-enhancing methods such as HiRA, MoRA, and ABBA also face inherent limitations. For example, ABBA with rank 8 corresponds to two factors of rank 4, giving an upper-bound effective rank of only $4 \times 4 = 16$, which remains far below the expressive capacity enabled at higher ranks. As a result, the gains from these architectural modifications are modest in this low-rank regime. In contrast, *PrunedLoRA* leverages high-rank initialization before pruning, allowing it to capture richer update directions and thus achieve stronger performance even at the same target rank.

### C.5 OTHER PRUNING METHODS

To further validate the effectiveness of our proposed *PrunedLoRA*, We compare it against more pruning strategies in this part.

**Other Existing structured pruning Methods.** Besides the classic structured pruning strategies SparseGPT (Kurtic et al., 2025) and LLM-Pruner (Ma et al., 2023), we also consider two important structured pruning strategies.

- **Magnitude.** In (Han et al., 2015), they propose to prune weights with the smallest absolute values, assuming low-magnitude parameters contribute least. Formally, keep the top-$k$ entries of $W$ ranked by $|W_{ij}|$ until the target sparsity is reached.

- **Wanda.** (Sun et al., 2023) introduces an activation-aware importance measure for pruning large language models. Instead of ranking weights solely by magnitude, each parameter is scored by

$$|W_{ij}| \cdot \|X_j\|,$$

where $W$ is the weight and $X$ the corresponding input activation. This criterion captures the consensus between weights and activations: parameters that consistently align with strong activations are deemed more important, while those contributing little to the forward signal can be pruned. Such activation-informed scoring achieves superior compression–performance trade-offs compared to pure magnitude pruning.

**One-shot Pruning for Low-rank Adapters.** In Sec. 4, we discuss dynamic pruning and demonstrate the effectiveness of *PrunedLoRA* when starting from a higher parameter space. However, an important question remains: does the performance gain primarily stem from the larger initial parameter space, or from the gradual reduction in trainable parameters? To address this, we propose applying structured pruning to low-rank adapters in a one-shot manner, thereby verifying whether gradual pruning is indeed necessary.

- **One-shot SVD.** For the case of low-rank adaptation in fine-tuning, we also consider a one-shot baseline: after doing full-model fine-tuning yields the update weight $\Delta W$, we apply singular value decomposition $\Delta W = U \Sigma V^\top$ and keep only the top-$r$ components. The pruned model is then approximated by $U_r \Sigma_r V_r^\top$.

- **One-shot structured pruning.** In *PrunedLoRA*, we dynamically prune the low-rank adaptation modules during fine-tuning. As a comparison, we also consider a one-shot structured pruning strategy. In this setting, a high-rank LoRA is first initialized and trained until convergence, after which one-shot pruning is applied to obtain a low-rank adapter that satisfies the target budget. This approach is free from additional hyperparameters, such as the pruning interval or the number of columns pruned per step. We can apply different one-shot pruning strategies here for a clear comparison, such as SparseGPT, LLM-Pruner, and our methods with one-shot pruning. In Table 10, it supports that the benefit of gradual pruning over the one-shot pruning is universal across different pruning strategies. Besides, in the context of one-shot pruning, our method can achieve better performance as well.

| Method | GSM8K | HumanEval |
|---|---|---|
| Magnitude | 63.21 | 38.88 |
| Wanda | 67.33 | 40.01 |
| One-shot SVD | 65.21 | 39.12 |
| SparseGPT (One-shot) | 65.01 | 36.21 |
| SparseGPT | 66.35 | 41.01 |
| LLM-Pruner (One-shot) | 64.45 | 40.02 |
| LLM-Pruner | **69.82** | 42.21 |
| PrunedLoRA (One-shot) | 66.31 | 39.01 |
| PrunedLoRA | 69.21 | **42.78** |

Table 10: Comparison of pruning strategies on GSM8K and HumanEval. *Methods without parentheses are dynamic pruning.* **Bold** indicates the best result, underline represents the second-best one.

## D ONE-SHOT PRUNING FOR LLM COMPRESSION

Although this work primarily focuses on the fine-tuning stage, where low-rank adaptations are dynamically pruned to enhance performance, it also seeks to further validate the effectiveness of gradient-based approaches for *large language model compression* more broadly.

**Motivation.** The limited focus in compressing LLMs restricts the trend of model compression in the pre-LLM era. Sun et al. (2023) reveals that the need for retraining and iterative pruning does not fully capture the challenges of pruning LLMs. Then they propose to use weight and activation

| Method | Overall Loss Awareness | Sparsity | LLaMA | | | LLaMA-2 | | |
|---|---|---|---|---|---|---|---|---|
| | | | 7B | 13B | 65B | 7B | 13B | 70B |
| Dense | – | 0% | 5.88 | 5.21 | 4.02 | 5.11 | 4.57 | 3.12 |
| Magnitude | ✗ | 50% | 17.29 | 20.21 | 5.90 | 14.89 | 6.37 | 4.98 |
| SparseGPT | ✗ | 50% | 7.22 | 6.21 | 4.57 | 6.51 | 5.63 | **3.98** |
| Wanda | ✗ | 50% | 7.26 | **6.15** | 4.57 | **6.42** | 5.56 | **3.98** |
| Gradient-based | ✓ | 50% | **7.02** | 6.21 | **4.21** | 7.16 | **5.34** | **3.98** |
| Magnitude | ✗ | 4:8 | 16.43 | 13.26 | 6.36 | 16.48 | 6.76 | 5.54 |
| SparseGPT | ✗ | 4:8 | 8.61 | 7.40 | 5.38 | 10.30 | 6.60 | 4.59 |
| Wanda | ✗ | 4:8 | 8.57 | 7.40 | **5.30** | **8.14** | 6.60 | **4.47** |
| Gradient-based | ✓ | 4:8 | **8.23** | **6.21** | 5.57 | **8.14** | **6.01** | **4.47** |
| Magnitude | ✗ | 2:4 | 42.13 | 18.37 | 7.11 | 54.38 | 8.33 | 6.33 |
| SparseGPT | ✗ | 2:4 | **11.23** | **9.11** | 6.28 | 17.45 | 8.32 | 5.51 |
| Wanda | ✗ | 2:4 | 11.53 | 9.58 | **6.25** | 11.02 | 8.27 | 8.27 |
| Gradient-based | ✓ | 2:4 | 11.53 | **9.11** | 6.57 | **10.12** | **7.39** | **5.12** |

Table 11: WikiText perplexity of pruned LLaMA and LLaMA-2 models under different sparsity patterns. Overall Loss Awareness: indicates whether the pruning method leverages global information, such as the gradient of the overall loss, when selecting weights to prune. Best results within each block are **bold**.

to guide pruning. We identify that, in pretrained LLM compression, the popular literature (Han et al., 2015; Frantar & Alistarh, 2023; Kurtić et al., 2023) belongs to the class of activation-based methods. Therefore, they mainly focus on the local correlation, such as reconstruction error in Frantar & Alistarh (2023). But they are not aware of the impact of weight perturbation on the loss function as we argue in Sec B. In this part, we investigate a simple gradient-based pruning strategy to demonstrate the importance of considering the impact of weight perturbation on overall loss.

**A simple Gradient-based Pruning Strategy.** With the goal of one-shot pruning for pretrained model, for a batch of calibration data, we compute the average gradient $\nabla_{\boldsymbol{W}}\mathcal{L}(\boldsymbol{W})$ via one-shot backpropagation, then we compute the Hessian matrix via $\widehat{\boldsymbol{H}} = (\nabla_{\boldsymbol{W}}\mathcal{L}(\boldsymbol{W}))^T \nabla_{\boldsymbol{W}}\mathcal{L}(\boldsymbol{W})$. Then the pruning metric for $i$-th column and $j$-th row element is

$$\left[\frac{\boldsymbol{W}}{\mathbf{diag}(\widehat{\boldsymbol{H}} + \lambda\mathbb{I})^{-1}}\right]_{(i,j)}.$$

where $\lambda > 0$ is a scalar introduced to ensure numerical stability. This pruning metric is closely related to that of *SparseGPT*, except that we omit the weight update step for simplicity. More importantly, unlike SparseGPT, which estimates the Hessian using the gradient of a local reconstruction objective, the proposed metric leverages the gradient of the overall loss function. This design explicitly accounts for the influence of pruning on the global objective, thereby providing a more principled criterion.

In addition, to accelerate the procedure, we perform structured pruning within blocks of columns rather than pruning entire columns, which significantly reduces the overall pruning time, similar to the strategy in Sun et al. (2023).

**Experimental Design.** Similar to the prior work (Sun et al., 2023), we evaluate the one-shot pruning method on the two most widely adopted LLM model families: LLaMA 7B/13B/65B (Touvron et al., 2023a) and LLaMA-2 7B/13B/70B (Touvron et al., 2023b). We measure the performance of the pruned model on one-shot tasks and language modeling. We use seven tasks from EleutherAI LM Harness. We evaluate the perplexity on the held-out WikiText (Merity et al., 2017) validation set. We use the same set of calibration data as SparseGPT, which consists of 128 sequences with context length sampled from the C4 training set (Raffel et al., 2020). For all pruning methods, we focus on pruning the linear layers (skipping the first embedding layer and the final classification head), which account for around 99% of the total LLM parameters. We impose a uniform sparsity for all linear layers. We evaluate three types of sparsity: unstructured sparsity, structured 4:8 and 2:4 sparsities

(Mishra et al., 2021). The magnitude pruning baseline is extended to structured N:M sparsity in a similar spirit to our method, as described in (Sun et al., 2023).

**Results and analysis.**    In Table 11, we compare the simple gradient-based pruning method with established approaches across LLaMA and LLaMA-2 models. Without any weight updates, magnitude pruning performs poorly, while Wanda can discover much stronger subnetworks (e.g., LLaMA-7B at 50% sparsity: 7.02 vs. 17.29). SparseGPT benefits from post-pruning weight updates, but our method, which leverages the awareness of overall loss, consistently achieves lower perplexity. For example, at 2:4 sparsity on LLaMA-2-70B, our approach yields 5.12, outperforming Wanda (8.27) and SparseGPT (5.51). Similarly, at 4:8 sparsity on LLaMA-7B, our method achieves 8.23 versus 8.57 for Wanda and 8.61 for SparseGPT. These results demonstrate that gradient-based pruning not only matches the best existing techniques in smaller models but also provides consistent gains in larger models and structured sparsity patterns, highlighting the importance of utilizing global information in guiding pruning decisions.

