# OpenReview forum: "PrunedLoRA: Robust Gradient-Based Structured Pruning for Low-rank Adaptation in Fine-tuning"
_ICLR.cc/2026/Conference — Submitted to ICLR 2026_

### Official Review · Reviewer_Ki12 · 2025-10-25

**Soundness:** 2
**Presentation:** 2
**Contribution:** 2
**Rating:** 2
**Confidence:** 4

**Summary:**

This paper proposes PrunedLoRA, a method that starts from a high-rank LoRA adapter and progressively prunes it using gradient- and Hessian-based criteria to achieve a compact final rank. The goal is to leverage higher expressivity during training while maintaining LoRA’s efficiency at inference. Experiments on math, code, and GLUE benchmarks show consistent gains over LoRA, DoRA, and AdaLoRA and other structured pruning methods, especially when starting from large initial ranks.

**Strengths:**

- The major idea in the paper is intuitive and clearly explained.


- The method is sound and evaluated on multiple tasks.


- The paper is clearly written and easy to follow.

**Weaknesses:**

`Peak memory will be significantly higher than LoRA, with no measurements reported.`

The peak memory usage in the paper will be significantly higher than pure LoRA (and other baselines at the base rank 64). For instance, when PrunedLoRA starts with rank 512, the memory required for parameters, gradients, and optimizer states (all of which occupy a large portion of memory) will increase roughly 8 times. The paper completely neglects mentioning any memory comparisons, which is very important in this case, since peak memory usage is a key metric for parameter-efficient fine-tuning.


`Missing comparisons to strong high-expressivity PEFT baselines.`

The paper does not compare against strong and relevant baselines that focus on high-expressivity and high effective rank PEFT methods, such as HiRA (https://openreview.net/forum?id=TwJrTz9cRS), MoRA (https://arxiv.org/abs/2405.12130), and ABBA (https://arxiv.org/abs/2505.14238). These methods are designed to achieve similar goals while maintaining efficiency, and excluding them makes it difficult to assess how much improvement actually comes from the proposed approach.


`Misaligned focus on inference efficiency instead of training-time memory.`

The authors mention that they optimize for the final inference efficiency of LoRA. This seems misplaced, PEFT methods generally optimize for training-time memory budgets, not for the small differences in storage caused by lower-rank adapters. Disk space is far cheaper than compute memory, and in many cases, adapters are even merged during inference, making storage savings largely irrelevant in practice.


`Can you compare against pure high-rank LoRA?`

It would be very helpful if the authors showed performance comparisons against pure LoRA trained at higher ranks. This would help characterize how close the pruning method comes to pure high-rank LoRA, which should typically serve as the upper-bound performance baseline (“skyline”).

**Questions:**

Please refer to weaknesses.

---

> ### Author Response · Authors · 2025-11-22
>
> Thanks for the reviewer’s suggestions and insightful comments. We address the concerns as follows:
>
> -------------
>
> ### 1. Peak Memory Cost.
>
> > Peak memory will be significantly higher than LoRA, with no measurements reported.
>
> The memory footprint of storing a low-rank adaptation with rank 512 is approximately 8 times of LoRA with rank 64 as the reviewer pointed out. But it is still less than fully fine-tuning significantly. Following the ablution study in LoRA-Pro, we report the memory footprint and training time in Table 2 of the updated paper. We measure memory cost using 8 H100 GPUs with a batch size 1.  We make two key observations: (1) Even with a very high initial LoRA rank, such as 512, the peak memory consumption of PrunedLoRA remains substantially lower than that of full
> fine-tuning. (2) The additional computation introduced by the pruning procedure incurs only a mild
> overhead beyond the standard LoRA forward and backward passes. Empirically, LoRA with rank
> 64 requires 2h 28 min, while PrunedLoRA (target rank 8, initial rank 64) completes in 2h 29 min
> under the same setup. It demonstrates that the pruning step adds mild runtime cost.
>
>
> ----------
>
> ***Table 1*** Comparison of trainable parameter ratios, peak memory cost (per GPU on 8×H100 with FSDP), and training time across different fine-tuning methods.
>
>
>
> | Method                     | Rank | Before ($\%$) | After ($\%$) | Memory (GB)        | Training Time |
> |----------------------------|------|------------|-----------|--------------------|---------------|
> | Full FT                    | --   | 100.00     | 100.00    | $\sim$ 8 $\times$ 40G          | 4h 23min      |
> | LoRA                       | 8    | 0.11       | 0.11      | $\sim$ 8 $\times$17G          | 2h 27min      |
> | LoRA                       | 64   | 0.84       | 0.84      | $\sim$ 8 $\times$ 18G          | 2h 28min      |
> | DoRA                       | 64   | 0.89       | 0.89      | $\sim$ 8 $\times$ 18G          | 2h 34min      |
> | AdaLoRA                    | 64   | 0.84       | 0.84      | $\sim$ 8 $\times$ 19G          | 2h 41min      |
> | PrunedLoRA (init r = 64)   | 8    | 0.84       | 0.11      | $\sim$ 8 $\times$ 19G          | 2h 29min      |
> | PrunedLoRA (init r = 128)  | 64   | 1.68       | 0.84      | $\sim$ 8 $\times$ 20G          | 2h 31min      |
> | PrunedLoRA (init r = 256)  | 64   | 3.36       | 0.84      | $\sim$ 8 $\times$ 22G          | 2h 47min      |
> | PrunedLoRA (init r = 512)  | 64   | 6.71       | 0.84      | $\sim$ 8 $\times$ 28G          | 3h 23min      |
>
> -------------
>
> ## 2. Comparison to other PEFT methods.
>
> > Missing comparisons to strong high-expressivity PEFT baselines.
>
>
>
> We thank reviewer *Ki12* for suggesting strong additional PEFT baselines. Regarding expressivity in low-rank adaptation, HiRA [1], MoRA [2], and ABBA [3] are designed to preserve a higher upper bound on the rank of the update matrix. MoRA introduces a square matrix to enable high-rank updates, while HiRA and ABBA exploit properties of the Hadamard product to expand the effective rank. These methods explicitly modify the LoRA architecture to enhance expressivity. In contrast, PrunedLoRA approaches the problem from a model-compression perspective: it leverages early-stage over-parameterization by starting from a high rank and pruning down to a compact low-rank structure. Thus, the underlying motivation of our method is fundamentally different from these prior approaches. In response to the reviewer’s request, we further compare HiRA, MoRA, and ABBA with PrunedLoRA on GSM8K using the Llama-3-8B base model, following the experimental setup in Table 1 in the updated paper. As shown in Appendix C.4.2, PrunedLoRA continues to exhibit strong performance, particularly at the target rank of 8. Notably, in this narrow-rank regime, ABBA uses rank 4 for each component, yielding an effective upper-bound rank of 16, whereas PrunedLoRA begins with a substantially higher rank (such as 64) and then compresses to the target rank, allowing it to retain useful expressivity despite the final compact form.

---

> ### Author Response · Authors · 2025-11-22
>
> -----------
>
> ***Table 2.*** Comparison between PrunedLoRA and expressiveness-enhancing PEFT variants (HiRA, MoRA, ABBA) on GSM8K with target ranks 8 and 64 using the pretrained Llama-3-8B base model.
>
>
> | Method      | Target Rank          | GSM8K ($\%$)          |
> |-------------|-----------------------|---------------------|
> | **LoRA**    | 8                     | 65.27 ± 0.13        |
> |  **LoRA**           | 64                    | 69.21 ± 0.36        |
> | **HiRA**    | 8                     | 67.38 ± 0.09        |
> | **HiRA**            | 64                    | 72.01 ± 0.19        |
> | **MoRA**    | 8                     | 68.88 ± 0.11        |
> |  **MoRA**           | 64                    | 71.08 ± 0.21        |
> | **ABBA**    | 8                     | 67.78 ± 0.18        |
> |  **ABBA**           | 64                    | 72.23 ± 0.24        |
> | **PrunedLoRA** | 8   (init r = 64)  | 69.02 ± 0.12        |
> |  **PrunedLoRA**           | 64 (init r = 128)     | 71.16 ± 0.34        |
> |    **PrunedLoRA**         | 64 (init r = 256)     | 72.21 ± 0.45        |
> |    **PrunedLoRA**         | 64 (init r = 512)     | 74.88 ± 0.42        |
>
> -------------
>
>
> ## 3. Inference efficiency and Training-time memory.
>
> > Misaligned focus on inference efficiency instead of training-time memory.
>
>
>  - We would like to clarify that our method does not solely focus on inference efficiency. Our ultimate goal is to narrow the gap between LoRA and full fine-tuning in the empirical performance. We will update a more clear statement in the section of Introduction.  We initialize LoRA with a high rank and dynamically prune it to be low-rank naturally help to achieve training efficiency compared with full fine-tuning as we analyzed in Section 3.3. Moreover, in the updated manuscript, we include the measurement of memory cost following the ablation study in LoRA-Pro (See Table 2). We make two key observations: (1) Even with a very high initial LoRA rank—such as 512—the peak memory consumption of PrunedLoRA remains substantially lower than that of full fine-tuning. (2) The additional computation introduced by the pruning procedure incurs only a mild overhead beyond the standard LoRA forward and backward passes. Empirically, LoRA with rank
> 64 requires 2h 28 min, while PrunedLoRA (target rank 8, initial rank 64) completes in 2h 29 min under the same setup. It demonstrates that the pruning step adds a mild runtime cost. Overall, our method is not motivated by storage reduction; rather, it improves model expressiveness while retaining fine-tuning efficiency, thereby bridging the performance gap with full fine-tuning.
>
>
>
>
>
> ## 4. Compare against pure high-rank LoRA.
>
>  > Can you compare against pure high-rank LoRA?
>
>
> We appreciate the reviewer for raising this valuable point. Higher-rank LoRA models indeed provide a natural upper-bound (“skyline”) baseline for evaluating the effectiveness of pruning-based approaches. To address this, we include additional comparisons against pure LoRA trained at higher ranks. As shown in the table below (or See Table 4 in the updated manuscript), PrunedLoRA—with a fixed target rank of 64 but initialized from higher ranks—closely tracks the performance of high-rank LoRA while using significantly fewer parameters during inference. In particular, initializing PrunedLoRA at rank 512 yields performance that is nearly identical to LoRA-512, demonstrating that structured pruning can recover most of the benefits of high-rank adaptation while maintaining a compact, low-rank representation at convergence.
>
> ------------
>
> ***Table 3***: Comparison between LoRA and PrunedLoRA at different target and initial ranks.
>
>
> | Method                    | Target Rank | GSM8K ↑ | HumanEval ↑ |
> |---------------------------|-------------|--------:|------------:|
> | LoRA                      | 128         |  72.51  |     44.82   |
> | LoRA                      | 256         |  73.83  |     46.48   |
> | LoRA                      | 512         |  74.98  |     48.49   |
> | PrunedLoRA (init r = 128) | 64          |  71.16  |     44.32   |
> | PrunedLoRA (init r = 256) | 64          |  72.21  |     46.21   |
> | PrunedLoRA (init r = 512) | 64          |  74.88  |     48.31   |
>
>
> --------
>
> The authors fully understand the demands on the reviewers’ time and sincerely appreciate their efforts in helping improve this work. The authors look forward to any further feedback the reviewers may have.
>
>
> --------
>
> [1] Huang Q, Ko T, Zhuang Z, et al. HiRA: Parameter-efficient hadamard high-rank adaptation for large language models. The Thirteenth International Conference on Learning Representations, 2025.
>
> [2] Jiang T, Huang S, Luo S, et al. Mora: High-rank updating for parameter-efficient fine-tuning. arXiv, 2024.
>
> [3] Singhal R, Ponkshe K, Vartak R, et al. ABBA: Highly Expressive Hadamard Product Adaptation for Large Language Models. arXiv, 2025.

---

> ### Comment · Reviewer_Ki12 · 2025-11-25
>
> After carefully reading the authors’ responses and the discussions with other reviewers, I have decided to keep my original score of 2. I remain unconvinced by the core motivation of the paper, and I do not believe that the increased training-time memory requirements are justified by the relatively modest improvements in inference efficiency.

---

> > ### Author Response · Authors · 2025-11-26
> >
> > We thanks for the reviewer's feedback in identifying potential concerns in our manuscript, which we have diligently addressed in our first-round response:
> >
> >  -  The peak memory cost has been reported (See Table 2 in the updated manuscript). It shows that the peak memory cost of our method in training time is much less than full fine-tuning.
> >
> >  - We carefully compare with other high-expressivity methods as the reviewer request to highlight the benefit of our methods (when the target rank is small).
> >
> >  - We further clarify our methods take training and inference efficiency into account together. The training efficiency has been validated by our empirical experiment and the inference efficiency is further interpreted from practical deployment.
> >
> >  - We report the performance of our method and the high rank LoRA, which shows that the power of pruning as we proposed can well preserve the high representative capacity.
> >
> >
> > For the second-round suggestions from the reviewer, the reviewer raise two concern and we would like to make a clear explanation here.
> >
> >
> > > Core motivation of our work
> >
> > Increasing the LoRA rank (e.g. 512) significantly narrows the performance gap with full fine-tuning. This has been empirically verified not only in our supervised fine-tuning experiments but also demonstrated in reinforcement learning settings, where a rank of 512 showed the same high representational capacity as FFT [1]. To effectively narrow this empirical performance gap when using the ***standard, low-rank regime (e.g., $r=8$)*** with full fine-tuning, we propose an initialization strategy: initializing LoRA with a high initial rank (still considerably less than the full matrix dimension) and then pruning it down to the desired low target rank. The pruning strategy would introduce modest training cost overhead and produce the low-rank adapters with the same architecture as LoRA.
> >
> >
> > > Further justification for the training memory cost and inference efficiency.
> >
> > We respectfully disagree that "the relatively modest improvements in inference efficiency". From the aspect of practical deployment, given that a model is trained only once (or a few times for tuning) but is used for inference indefinitely by users, empirical applications typically prioritize minimizing inference costs. Therefore, improving inference efficiency is significant and meaningful. Besides, as our goal is to narrow the gap between LoRA with small ranks and full fine-tuning, producing the small low-rank adapters from our method will keep the consistent inference efficiency as standard LoRA with small target rank.
> >
> >
> >  [1] Schulman, John and Thinking Machines Lab, "LoRA Without Regret", Thinking Machines Lab: Connectionism, Sep 2025.

---

> > > ### Comment · Reviewer_Ki12 · 2025-11-28
> > >
> > > Thank you for the detailed rebuttal.
> > >
> > > I am still not fully convinced by the authors’ core motivation and the arguments regarding inference efficiency, for the following reason. In many scenarios, one can simply merge the adapter weights into the full model at inference time, resulting in identical efficiency regardless of the LoRA rank. This shifts the bottleneck to training compute, which will naturally be higher for PrunedLoRA. The method could be useful in settings where adapter storage is the primary concern, although this raises a separate set of considerations and would require a different set of baselines.
> > >
> > > That said, I respect the authors’ perspective, as is appropriate in scientific discourse. Taking this into account, along with the substantial effort put into the rebuttal, I will increase my score to 4.
> > >
> > > (PS: The OpenReview edit button is currently not working due to a bug, so I will update the score once the issue is resolved.)

---

> > > > ### Author Response · Authors · 2025-11-28
> > > >
> > > > We sincerely appreciate the reviewer for ***raising the score and further recognizing our work in the rebuttal***. We would like to provide additional clarification regarding our motivation and our statements about inference efficiency, as follows:
> > > >
> > > > We have strengthened the explanation of the core motivation by highlighting ***the empirical observation*** behind our design and ***our goal of narrowing the performance gap*** between standard low-rank adaptation (e.g., rank-8 LoRA) and full fine-tuning. As shown in Table 4 of the updated manuscirpt, the empirical comparison between PrunedLoRA and high-rank LoRA further validates this motivation.
> > > >
> > > > Regarding inference efficiency, we acknowledge that the wording in the Introduction may have unintentionally suggested that inference efficiency is our ultimate objective. This is not the case, as our primary motivation is described above. We have revised the relevant statements for greater clarity. We also understand the reviewer’s point that LoRA does not incur inference latency overhead after merging, and thus high rank may seem harmless. However, we would like to emphasize that in multi-task or multi-domain settings, it is necessary to store separate low-rank adapters for each task. In such scenarios, increasing the rank directly increases the memory footprint, making high-rank LoRA less desirable.

---

### Official Review · Reviewer_THbS · 2025-10-28

**Soundness:** 3
**Presentation:** 3
**Contribution:** 4
**Rating:** 6
**Confidence:** 4

**Summary:**

The paper proposes the PrunedLoRA framework, which starts from a high-rank (over-parameterized) LoRA initialization and dynamically prunes it during fine-tuning using a novel, gradient-based structured pruning strategy . The main contributions include: (1) The PrunedLoRA framework; (2) A theoretical analysis demonstrating that gradient-based pruning is more robust to weight perturbations than activation-based pruning; (3) An interpretation of a second-order pruning metric ("saliency"). Experimental results show that PrunedLoRA (especially with a high initial rank, e.g., r=512) matches or exceeds the performance of FFT on math, code, and NLU tasks, and outperforms standard LoRA and other pruning baselines.

**Strengths:**

1. Strong Empirical Performance: On key benchmarks, PrunedLoRA (with r=512) matches or even exceeds the performance of Full Fine-Tuning, while significantly outperforming standard LoRA and other pruning baselines (Table 1, 3).
2. Novel Theoretical Analysis: The paper is the first to theoretically demonstrate (Sec 3.2, Prop. 1 & 2) that the proposed gradient-based pruning method has stronger robustness (error is independent of module magnitude) under weight perturbations compared to activation-based methods.

**Weaknesses:**

1. Insufficient Justification for Hessian Approximation: Approximating the Hessian in Eq. 5 is a very strong assumption, but the paper addresses it with only a single sentence, without discussing its impact on the "optimality" of the derivation.
2. Gap Between Theory and Practice: The theoretical advantages of robustness claimed in Proposition 1 & 2 (e.g., "magnitude dependency") are not directly verified experimentally, weakening the link to the empirical performance gains.
3. Missing Computational Cost Analysis: The paper fails to clearly isolate the cost of the pruning algorithm itself (Table 2) and ignores the significant O(r^3) cost from the high initial rank r in its complexity analysis. The pruning schedule for the best-performing models is also missing.

**Questions:**

1. Your theoretical analysis (Prop. 1 & 2) presents a novel claim that gradient-based pruning is more robust than activation-based pruning. However, the paper lacks direct empirical validation of this specific claim. Could you provide a more convincing argument or (if feasible) preliminary experimental evidence to demonstrate that this specific theoretical robustness, and not just the general concept of "pruning from over-parameterization," is what drives PrunedLoRA's superior performance over baselines like SparseGPT?
2. You report in Table 2 that PrunedLoRA (r=512) takes ~53 minutes longer than standard LoRA. However, it is unclear how much of this 53-minute overhead comes from  the inherent cost of performing forward/backward passes at a higher rank (r=512) versus the pruning steps themselves. Could you provide a more detailed cost breakdown, specifically stating the total wall-clock time (or percentage of total time) consumed by the pruning algorithm (Alg. 1) itself during the r=512 run?

---

> ### Author Response · Authors · 2025-11-22
>
> Thanks for the reviewer’s suggestions and insightful comments. We address the concerns as follows:
>
> -------------
>
> ## 1. Approximation the Hessian.
>
> > Insufficient Justification for Hessian Approximation: Approximating the Hessian in Eq. 5 is a very strong assumption, but the paper addresses it with only a single sentence, without discussing its impact on the "optimality" of the derivation.
>
>  - We acknowledge that approximating the Hessian in Eq.~5 is a necessary assumption, but it is also a standard and widely adopted practice in second-order pruning methods (e.g., [1,2]). Importantly, computing the complete Hessian matrix in LLM is very difficult since it is easy to be out of memory. Our goal is not to claim exact optimality under the true Hessian, but to construct a tractable and reliable surrogate objective that enables efficient column selection while preserving the dominant curvature directions of $W$. To analysis the properties of approximating Hessian via outer product of gradient, prior work [3] has studied Hessian approximation using the empirical Fisher information in the context of natural gradient descent, offering further justification for such approximations. A rigorous analysis of the optimality gap induced by Hessian approximation is orthogonal to our main contribution and represents a promising direction for future work.
>
>
> ## 2. Gap between Theory and Practice for Propositions 1 and 2.
>
>  - Thanks to the reviewer for recognizing the novelty in distinguishing gradient-based pruning from activation-based pruning. Although the original paper did not include empirical evidence directly validating the robustness suggested by our theoretical analysis, our experiments already demonstrate the benefits of pruning while being aware of the global loss landscape. To further substantiate our theory, we have added ***a synthetic study using a simple self-attention model trained on a linear regression task*** (Appendix B.2). The purpose of this experiment is to directly compare how activation-based and gradient-based pruning influence the change in total loss under the same perturbation precision under different pruning criteria. The results show that activation-based pruning leads to a significantly larger increase in loss, as it cannot capture the components of the weight matrix that lie outside the activation reconstruction error. In contrast, gradient-based pruning—by explicitly accounting for the global loss function—produces much more stable perturbations, consistent with our theoretical prediction.
>
>
>
> ## 3. Computational Cost Analysis.
>
>
> > Missing Computational Cost Analysis: The paper fails to clearly isolate the cost of the pruning algorithm itself (Table 2) and ignores the significant $O(r^3)$ cost from the high initial rank r in its complexity analysis. The pruning schedule for the best-performing models is also missing.
>
>  - When the initial rank is 512, the computational complexity of our pruning step is $\mathcal{O}(r^3)$, whereas the complexity of the standard LoRA update is $\mathcal{O}(\max\\{m^2, n^2\\}\, r)$, where $m$ and $n$ denote the input and output dimensions of the weight matrix $W$. This indicates that, at such a high initial rank, the pruning cost is asymptotically lower than the regular LoRA forward--backward computation. To empirically validate this observation, we isolate and measure the
> wall-clock time required solely for the pruning procedure with initial
> rank $r = 512$ on the GSM8K benchmark. The time for conducting our pruning strategy is
> approximately ***13 minutes*** (6.40$\\\%$ in overall training time), demonstrating that the additional cost introduced by our second-order pruning strategy is modest in practice.

---

> ### Author Response · Authors · 2025-11-22
>
> ## 4. The proposed pruning schedule
>
> > The pruning schedule for the best-performing models is also missing.
>
>  - As shown in Appendix D.5, we compare Algorithm 2 with several alternative pruning strategies, including one-shot pruning (One-shot pruning or one-shot structured pruning) and more advanced structured pruning baselines (Magnitude and Wanda). Across all evaluated settings, Algorithm 2 consistently achieves the best performance. Based on these empirical results, we recommend a dynamic pruning schedule in which two pruning indices are added by Algorithm 2 every 10 training steps, which we find to provide a stable balance between pruning aggressiveness and model accuracy.
>
> ---------------
>
> The authors fully understand the demands on the reviewers’ time and sincerely appreciate their efforts in helping improve this work. The authors look forward to any further feedback the reviewers may have.
>
> ---------------
>
> [1] Singh S P, Alistarh D. Woodfisher: Efficient second-order approximation for neural network compression. Advances in Neural Information Processing Systems, 2020.
>
> [2] Kurtic E, Campos D, Nguyen T, et al. The Optimal BERT Surgeon: Scalable and Accurate Second-Order Pruning for Large Language Models. Proceedings of the 2022 Conference on Empirical Methods in Natural Language Processing, 2022.
>
> [3] Martens J. New insights and perspectives on the natural gradient method. Journal of Machine Learning Research, 2020.

---

> ### Author Response · Authors · 2025-11-27
>
> Dear Reviewer THbS,
>
> We sincerely appreciate the time and effort you have devoted to reviewing our manuscript and providing valuable suggestions and comments. We are encouraged that you found ***the strong empirical performance*** and ***novel theoretical analysis*** in our work to be noteworthy and ***your positive rating feedback***.
>
> We are also grateful for your feedback in identifying potential concerns. In our response, we have carefully addressed these points by providing a detailed justification for ***the Hessian approximation, adding synthetic experiments to validate the theory, and reporting the pruning computation time***. As the discussion period deadline approaches, we would greatly appreciate any further comments you may have, as we are eager to clarify and resolve any remaining issues.
>
> We truly appreciate the significant time and care you have devoted to reviewing our work. Your feedback has been invaluable in helping us further strengthen the paper.

---

### Official Review · Reviewer_ZZ6d · 2025-10-29

**Soundness:** 2
**Presentation:** 3
**Contribution:** 2
**Rating:** 2
**Confidence:** 3

**Summary:**

PrunedLoRA initializes LoRA with a high rank, then applies gradient/Hessian-guided structured pruning (column/row) with a closed-form compensation step to mitigate pruning-induced loss increase. The paper also provides theory suggesting gradient-based pruning is more loss-robust than activation-based criteria, and it reports consistent gains over LoRA/DoRA/AdaLoRA and structured-pruning baselines on GSM8K, HumanEval, and a GLUE subset.

**Strengths:**

1. **Clear and practical recipe**. High-rank initialization followed by structured pruning with second-order (gradient–Hessian) signals and closed-form updates is well-motivated and easy to implement conceptually.
2. **Consistent empirical improvements**. Across GSM8K, HumanEval, and GLUE (subset), PrunedLoRA narrows the gap to full FT and outperforms strong PEFT and pruning baselines under various sparsity/rank regimes

**Weaknesses:**

1. Motivation not fully compelling. The observation that "higher rank helps does not by itself necessitate pruning inside LoRA. Additionally, what concrete advantages does this scheme offer over alternatives? For instance, why not adopt AdaLoRA’s rank allocation, or directly learn a structurally sparse $\Delta W$? The paper should articulate a clearer conceptual and empirical rationale relative to these design points.

2. Narrow rank regime. Experiments primarily use relatively large ranks ($r\in[64,256]$ for initialization and prune to 64), whereas practical deployments often push to very small ranks ($r=1-8$). The original LoRA paper reports competitive results at $r=1-8$ on some tasks; please include systematic results and stability curves for $r\in\{1,2,4,8\}$. (Figures do explore small targets in ablations, but a consolidated, head-to-head comparison versus baselines at these tiny ranks would strengthen the case.)

3. Training cost vs. inference savings trade-off is under-argued.
The claimed inference-time storage benefit comes at the price of higher training overhead (start high-rank + second-order mask search + compensation). In many practical settings, training memory/compute, not inference storage, is the binding constraint.

4. Other experimental settings. See Questions  below.

**Questions:**

1. **Learning-rate search range**. In the appendix you state: "we perform grid search over $\{1e-5, 5e-6, 1e-6 \}$" This range is extremely low for LoRA; in practice, many LoRA setups peak around $[5e-5, 2e-4]$ (See [1]). Using such low LRs raises fairness concerns, especially because PrunedLoRA may benefit from larger effective updates early on due to higher initial rank. Please expand the grid to include standard LoRA ranges and report the best.

2. **Scaling factor $\frac{\alpha}{r}$**. LoRA updates are $W=W_0 +\frac{\alpha}{r} BA$. You mention grid search over learning rates and scaling factors for all methods, but I didn't find the description about scaling factors in Appendix C.1.

3. Is sequence length 128 enough for math or code tasks? And why you use a such small training batch size? Usually the value would be 32 in [1][2].


[1] Zhang, Y., Liu, F., & Chen, Y. (2025). LoRA-One: One-Step Full Gradient Could Suffice for Fine-Tuning Large Language Models, Provably and Efficiently. arXiv preprint arXiv:2502.01235.

[2] Wang, S., Yu, L., & Li, J. (2024). Lora-ga: Low-rank adaptation with gradient approximation. Advances in Neural Information Processing Systems, 37, 54905-54931.

---

> ### Author Response · Authors · 2025-11-22
>
> Thanks for the reviewer’s suggestions and insightful comments. We address the concerns as follows:
>
> ----------
>
> ## 1. Motivation for pruning inside LoRA
>
> > Motivation not fully compelling. The observation that "higher rank helps does not by itself necessitate pruning inside LoRA. Additionally, what concrete advantages does this scheme offer over alternatives? For instance, why not adopt AdaLoRA’s rank allocation, or directly learn a structurally sparse? The paper should articulate a clearer conceptual and empirical rationale relative to these design points.
>
>
>  - Pruning high ranks into low-rank adapters servers as a promising solution to narrow the gap between low-rank adaptation and full fine-tuning. It would help to mitigate the low-rank constraint and further improve the performance when optimizing LoRA and its variants during training (See Table 1 and 7). For the alternative strategies, as the reviewer mention: 1) ***AdaLoRA*** aims to allocate the parameter budget adaptively according to the importance of modules to improve the performance of parameter-efficient fine-tuning. Even though the rank in AdaLoRA can be dynamically allocated, the initial rank is 1.5 times of the final rank as they claimed (See Section 3.3 in AdaLoRA). Therefore, all parameter is always restricted into low-rank spaces. Our methods aim to keep the initial rank very high, and prune the unimportant ranks to output the low-rank adapters for efficient inference. The code idea from AdaLoRA is resource allocation and the key idea of our method is model compression. 2) ***Directly learning a structurally sparse update $\Delta W$*** is indeed feasible using standard pruning strategies, and such structured sparsity can provide inference-time efficiency when the weight pattern is hardware-friendly. We also explore applying SVD to $\Delta W$ to obtain a low-rank adapter as well (see one-shot SVD in Appendix D.5). However, both approaches suffer from a fundamental limitation: during training, one must still maintain the \emph{dense} update matrix $\Delta W$ together with its optimizer states (e.g., AdamW moments). As a result, the memory footprint and computational overhead are comparable to—or even exceed—those of full fine-tuning. This significantly limits their practicality.
>
> ## 2. Narrow rank regime.
>
> > Experiments primarily use relatively large ranks ($r \in [64, 256]$ for initialization and pruning to $64$), whereas practical deployments often push to very small ranks ($r = 1$--$8$).
> The original LoRA paper reports competitive results at $r = 1$--$8$ on some tasks; please include systematic results and stability curves for $r \in \{1, 2, 4, 8\}$.
> (Figures do explore small targets in ablations, but a consolidated, head-to-head comparison versus baselines at these tiny ranks would strengthen the case.)
>
>  - To provide a direct, head-to-head evaluation against the baselines as the reviewer suggested, we conducted additional experiments with a fixed target rank of 8. Table 1 in the updated manuscript reports results on dialogue, math, and code-generation tasks using the Llama-3 8B base model. We compare PrunedLoRA with vanilla LoRA and several representative variants AdaLoRA and DoRA. For all pruning-based methods, the initial rank is set to 64. Across all tasks, PrunedLoRA consistently outperforms standard LoRA with rank 8 as well as the other baselines, demonstrating that pruning from a high-rank initialization remains highly beneficial even in the low-rank regime. In addition, Appendix~C.4 further compares PrunedLoRA with advanced variants of LoRA such as LoRA+, LoRA-GA, LoRA-Pro, AltLoRA, HiRA, MoRA, and ABBA at the same target rank of 8 (See Table 8 and Table 9). These results further show the superior performance of our method in the small-rank setting.
>
> ---------
>
> ***Table 1***: Performance comparison of fine-tuning and pruning baselines on MT-bench, GSM8K and HumanEval benchmarks for Llama-3-8B-Base Model.
>
> | Method        | Target Rank | MT-Bench ↑        | GSM8K ↑             | HumanEval ↑          |
> |--------------|-------------|-------------------|---------------------|----------------------|
> | PreTrain     | --          | 5.89 ± 0.04       | 51.34 ± 1.38        | 34.21 ± 0.23         |
> | LoRA         | 8           | 6.01 ± 0.05       | 65.27 ± 0.13        | 39.23 ± 0.78         |
> | DoRA         | 8           | 6.07 ± 0.02       | 67.08 ± 0.31        | 41.28 ± 0.39         |
> | AdaLoRA      | 8           | 6.08 ± 0.03       | 71.24 ± 1.32        | 41.88 ± 1.15         |
> | SparseGPT    | 8           | 6.09 ± 0.02       | 67.28 ± 0.29        | 41.43 ± 0.28         |
> | LLM-Pruner   | 8           | 6.09 ± 0.03       | 69.88 ± 0.35        | 42.25 ± 0.32         |
> | PrunedLoRA   | 8           | 6.14 ± 0.06       | 69.02 ± 0.53        | 42.32 ± 0.33         |
>
> ----------

---

> ### Author Response · Authors · 2025-11-22
>
> ## 3. Training cost vs. inference saving.
>
> > Training cost vs. inference savings trade-off is under-argued. The claimed inference-time storage benefit comes at the price of higher training overhead (start high-rank + second-order mask search + compensation). In many practical settings, training memory/compute, not inference storage, is the binding constraint.
>
>
>  - The reviewer’s statement that our method is merely trading higher training cost for inference efficiency may not be accurate. Our primary goal is twofold: (i) to narrow the empirical performance gap between low-rank adaptation and full fine-tuning, and (ii) to obtain low-rank adapters with training and inference efficiency. We have clarified and emphasized this motivation in the introduction section. To validate our method's training efficiency versus full fine-tuning,  Table 2 in the updated manuscript provides a detailed comparison of peak memory usage and training time across PrunedLoRA and several baselines. As shown in Table 2, PrunedLoRA remains substantially more efficient than full fine-tuning in practice. In particular, full fine-tuning requires approximately $8\times40$G of memory and over 4 hours of training time, while PrunedLoRA, even when initialized from a very large rank such as $512$, uses significantly less memory (approximately $8\times28$G) and still finishes within 3h 23min. More importantly, when initialized with moderate ranks ($64$--$256$), PrunedLoRA achieves memory usage comparable to LoRA/DoRA/AdaLoRA while maintaining nearly identical training time (2h 29min--2h 47min). This demonstrates that introducing second-order pruning incurs only a mild computational overhead on top of standard LoRA training. Finally, while we agree that training efficiency is an important consideration, obtaining lightweight update modules for inference is equally crucial: training is typically performed once, whereas inference is executed many times in real-world deployments. Our design explicitly aware this trade-off by improving low-rank performance without incurring the prohibitive cost of full fine-tuning.
>
> -------
>
> ***Table 2*** Comparison of trainable parameter ratios, peak memory cost (per GPU on 8×H100 with FSDP), and training time across different fine-tuning methods.
>
> | Method                     | Rank | Before ($\%$) | After ($\%$) | Memory (GB)        | Training Time |
> |----------------------------|------|------------|-----------|--------------------|---------------|
> | Full FT                    | --   | 100.00     | 100.00    | $\sim$ 8 $\times$ 40G          | 4h 23min      |
> | LoRA                       | 8    | 0.11       | 0.11      | $\sim$ 8 $\times$17G          | 2h 27min      |
> | LoRA                       | 64   | 0.84       | 0.84      | $\sim$ 8 $\times$ 18G          | 2h 28min      |
> | DoRA                       | 64   | 0.89       | 0.89      | $\sim$ 8 $\times$ 18G          | 2h 34min      |
> | AdaLoRA                    | 64   | 0.84       | 0.84      | $\sim$ 8 $\times$ 19G          | 2h 41min      |
> | PrunedLoRA (init r = 64)   | 8    | 0.84       | 0.11      | $\sim$ 8 $\times$ 19G          | 2h 29min      |
> | PrunedLoRA (init r = 128)  | 64   | 1.68       | 0.84      | $\sim$ 8 $\times$ 20G          | 2h 31min      |
> | PrunedLoRA (init r = 256)  | 64   | 3.36       | 0.84      | $\sim$ 8 $\times$ 22G          | 2h 47min      |
> | PrunedLoRA (init r = 512)  | 64   | 6.71       | 0.84      | $\sim$ 8 $\times$ 28G          | 3h 23min      |
>
> ------------
>
>
> ## 4. Learning Rate search range.
>
>
> > In the appendix you state: ``we perform grid search over $1\mathrm{e}{-5}, 5\mathrm{e}{-6}, 1\mathrm{e}{-6}$.'' This range is extremely low for LoRA; in practice, many LoRA setups peak around $[5\mathrm{e}{-5}, 2\mathrm{e}{-4}]$. Using such low learning rates raises fairness concerns, especially because PrunedLoRA may benefit from larger effective updates early on due to its higher initial rank.
> Please expand the grid to include standard LoRA ranges and report the best.
>
>
>
> -  We acknowledge that our initial learning-rate search range was narrower than the default configurations used in LoRA-Pro and AltLoRA. In response, we have retrained the tasks of language generation with an expanded grid that includes the standard LoRA learning-rate range $\\{2e-4, 5e-5, 2e-5\\}$ and the sequence length 1024, and we now report the best-performing results in Table~1. The experiment design has been reported in Section 4.1 within the updated manuscript.
>
> ## 5. Scaling factor $\alpha / r$.
>
> > LoRA updates are $W = W_{0} + \frac{\alpha}{r} BA$.
> You mention grid search over learning rates and scaling factors for all methods, but I did not find the description of the scaling factors in Appendix~C.1.
>
> The selection of the hyperparameter $\alpha$ has been clarified and reported in Section C.2 of the updated manuscript. We perform a grid search over a large range and identify that $\alpha \in \\{r/2, r,2r\\}$ can achieve the better performance.

---

> ### Author Response · Authors · 2025-11-22
>
> ## 6. Sequence Length.
>
> > Is sequence length 128 enough for math or code tasks? And why you use a such small training batch size? Usually the value would be 32 in [1][2].
>
> We agree that a sequence length of 128 is relatively short. Accordingly, we have retrained all tasks of language generation using a sequence length of 1024 and a batch size of 32. The updated results as show below are presented in Table 1 of the new manuscript.
>
> -------------
> ***Table 3***: Performance comparison of fine-tuning and pruning baselines on MT-bench, GSM8K and HumanEval benchmarks for Llama-3-8B-Base Model.
>
>
> | Method        | Target Rank      | MT-Bench ↑        | GSM8K ↑             | HumanEval ↑          |
> |--------------|------------------|-------------------|---------------------|----------------------|
> | PreTrain     | --               | 5.89 ± 0.04       | 51.34 ± 1.38        | 34.21 ± 0.23         |
> | Full FT      | --               | 6.31 ± 0.03   | 73.48 ± 0.42 | 48.28 ± 0.03  |
> | LoRA         | 8                | 6.01 ± 0.05       | 65.27 ± 0.13        | 39.23 ± 0.78         |
> | LoRA         | 64               | 6.19 ± 0.03       | 69.21 ± 0.36        | 42.88 ± 0.34         |
> | DoRA         | 8                | 6.07 ± 0.02       | 67.08 ± 0.31        | 41.28 ± 0.39         |
> | DoRA         | 64               | 6.23 ± 0.03       | 70.43 ± 0.21        | 43.32 ± 0.29         |
> | AdaLoRA      | 8                | 6.08 ± 0.03       | 71.24 ± 1.32        | 41.88 ± 1.15         |
> | AdaLoRA      | 64               | 6.12 ± 0.08       | 71.45 ± 1.37        | 42.34 ± 1.41         |
> | SparseGPT    | 8                | 6.09 ± 0.02       | 67.28 ± 0.29        | 41.43 ± 0.28         |
> | SparseGPT    | 64               | 6.16 ± 0.02       | 69.71 ± 0.48        | 43.82 ± 0.39         |
> | LLM-Pruner   | 8                | 6.09 ± 0.03       | 69.88 ± 0.35        | 42.25 ± 0.32         |
> | LLM-Pruner   | 64               | 6.18 ± 0.03       | 70.88 ± 0.45        | 44.38 ± 0.12         |
> | PrunedLoRA   | 8                | 6.14 ± 0.06       | 69.02 ± 0.53        | 42.32 ± 0.33         |
> | PrunedLoRA   | 64               | 6.19 ± 0.04       | 71.16 ± 0.24        | 44.32 ± 0.11         |
> | PrunedLoRA   | 64 (init r = 256)| 6.23 ± 0.03       | 72.21 ± 0.45        | 46.21 ± 0.26         |
> | PrunedLoRA   | 64 (init r = 512)| 6.25 ± 0.06 | 74.88 ± 0.42    | 48.31 ± 0.24     |
> -------------
>
> The authors fully understand the demands on the reviewers’ time and sincerely appreciate their efforts in helping improve this work. The authors look forward to any further feedback the reviewers may have.

---

> > ### Comment · Reviewer_ZZ6d · 2025-11-26
> >
> > I appreciate the authors’ response, particularly their addition of experiments to address many issues in the initial version. However, I still have concerns regarding the core motivation of the paper (pruning high-rank LoRA to low-rank for better approximation of full fine-tuning) and the additional training overhead it introduces. Therefore, I raise my score to 4.

---

> > > ### Author Response · Authors · 2025-11-26
> > >
> > > We thanks for the reviewer’s feedback in identifying potential concerns in our manuscript, which we have
> > > diligently addressed in our first-round response:
> > >
> > > - We discuss the advantages of our methods over the alternatives (AdaLoRA and sparse $\Delta W$)as the reviewer proposed.
> > >
> > > - We conduct additional experiments with smaller target rank in LoRA showing that pruning from a high-rank initialization remains highly beneficial even in the small rank regime.
> > >
> > > - We report the peak memory cost in training in Table 2 of the updated manuscript, which demonstrates PrunedLoRA remains substantially more efficient than full fine-tuning.
> > >
> > > - We provide update experiment results with comparable hyperparameters like sequence length and learning rate.
> > >
> > > ------
> > >
> > > For the second-round suggestions from the reviewer, the reviewer raises two concern and we would like to make a clear explanation here.
> > >
> > > > the core motivation of the paper (pruning high-rank LoRA to low-rank for better approximation of full fine-tuning)
> > >
> > > Increasing the LoRA's rank (e.g. 512) significantly narrows the performance gap with full fine-tuning. This has been empirically verified not only in our supervised fine-tuning experiments but also demonstrated in reinforcement learning settings, where a rank of 512 showed the same high representational capacity as FFT [1]. To effectively narrow this empirical performance gap when using the ***standard, low-rank regime (e.g., r=8)*** with full fine-tuning, we propose an initialization strategy: inspired by model compression, initializing LoRA with a high initial rank (still considerably less than the full matrix dimension) and then pruning it down to the desired low target rank. Compared with other compression strategies (such as sparse $\Delta W$ the reviewer suggested, our methods would introduce modest training cost overhead and produce the low-rank adapters with the same architecture as LoRA. As shown in Table 4 of the updated manuscript, PrunedLoRA would preserve similar empirical performance as high-rank LoRA has. It further validate the effectiveness of our motivation.
> > >
> > >
> > >
> > >
> > > > the additional training overhead it introduces
> > >
> > > We acknowledge that our method introduces additional training overhead due to the second-order pruning step and high-rank initialization. However, as reported in Table~2 of the updated manuscript, both the time and memory overhead remain modest in practice.
> > >
> > > 1) **computation cost**. The pruning step adds only a small amount of post-training computation. For example, when using an initial rank of $r$ on the GSM8K benchmark, we separately measured the wall-clock time required ***exclusively*** for the pruning procedure. The total pruning time is approximately 13 minutes, accounting for only $6.40\%$ of the overall training time. This indicates that the extra cost introduced by our second-order pruning strategy is limited in practice.
> > >
> > > 2) **memory cost**. The peak memory usage during training remains close to that of standard LoRA and is significantly lower than that required for full fine-tuning. This shows that our approach retains the memory-efficiency benefits of low-rank adaptation despite the higher-rank initialization.
> > >
> > > Overall, the empirical analyses in Table~2 demonstrate that the training overhead of PrunedLoRA is mild and does not undermine its practical usability.
> > >
> > >
> > >
> > >  [1] Schulman, John and Thinking Machines Lab, "LoRA Without Regret", Thinking Machines Lab: Connectionism, Sep 2025.

---

### Official Review · Reviewer_5szM · 2025-11-03

**Soundness:** 2
**Presentation:** 2
**Contribution:** 2
**Rating:** 2
**Confidence:** 3

**Summary:**

In this paper, the author proposed PrunedLoRA, a gradient-based structured pruning framework for obtaining efficient low-rank adapters from over-parameterized spaces. And they provide a theoretically grounded strategy for structured adapter compression.
Experiments over NLU and NLG tasks validate PrunedLoRA‘s effectiveness.

**Strengths:**

-	The paper is well-written and the figures are clear.
-	The gradient-based pruning theory is well-established and interesting.

**Weaknesses:**

-	My primary concern is the reliability of the experimental results. I noticed significant differences in the experimental setup compared to prior works [1,2,3]. The authors should clarify why their setup and results diverge so substantially.
    - For instance, the sequence length T=128 is unusually short—typically, T=1024 is used.
    -	The results in Table 1 also appear much weaker than those in [1], where the rank was only 8, far lower than the pre-pruning rank of 128 and post-pruning rank of 64 in Table 1.
    -	Work [1] trained for only 3,000 steps, whereas this study trained for 5,000 steps. Logically, the same method should yield significantly better results than [1].
-	Additional ablation studies are needed:
    -	MT-Bench results should be included in Table 1.
    -	Comparisons with newer baselines like LoRA-GA, LoRA-Pro, AltLoRA, and LoRA+ are missing—the current baselines are outdated.
    -	Need to include the results of Full Fine-Tuning using Algorithm 2.
    -	Table 2 should also report GPU memory usage during training.


[1] AltLoRA: Towards Better Gradient Approximation in Low-Rank Adaptation with Alternating Projections. In ArXiv 25.

[2] LoRA-Pro: Are Low-Rank Adapters Properly Optimized?. In ICLR 25.

[3] LoRA-GA: Low-Rank Adaptation with Gradient Approximation. In NeurIPS 24.

**Questions:**

See above.

---

> ### Author Response · Authors · 2025-11-22
>
> Thanks for the reviewer's suggestions and insightful comments. We address the concerns as follows:
>
> ------
>
>
> ## 1, Differences in the Experimental Setup
>
>
> > For instance, the sequence length T=128 is unusually short—typically, T=1024 is used.
>
> - We acknowledge that the sequence length used in our original experimental setup was not fully aligned with prior works, as the reviewer mentioned. To ensure a fair comparison, we have retrained all experiments of language generation tasks (dialogue generation, math and code tasks) using the standard configuration (e.g., sequence length $T = 1024$), and the updated results are now reported in the Experiments section (see Table 1 in the updated manuscript).
>
>
> > The results in Table 1 also appear much weaker than those in [1], where the rank was only 8, far lower than the pre-pruning rank of 128 and post-pruning rank of 64 in Table 1.
>
> - The results in our original Table 1 were produced using *Llama-3-8B*, whereas AltLoRA [1] reports results on Llama-3.1-8B, which possesses stronger pretrained capabilities. Compared with *Llama-3-8B*, *Llama-3.1-8B* introduces enhanced multilingual support and extends the context window from approximately 8K to 128K tokens, enabling more robust cross-lingual performance and long-document reasoning. This discrepancy in the base model partly explains the weaker performance we initially observed at lower LoRA ranks. In the revised manuscript, we have updated Table 1 using the corrected sequence length on *Llama-3-8B*, and the new results show that increasing the LoRA rank effectively narrows the performance gap.
>
>
> --------
>
> ***Table 1***: Performance comparison of fine-tuning and pruning baselines on MT-bench, GSM8K and HumanEval benchmarks for Llama-3-8B-Base Model.
>
>
> | Method        | Target Rank      | MT-Bench ↑        | GSM8K ↑             | HumanEval ↑          |
> |--------------|------------------|-------------------|---------------------|----------------------|
> | PreTrain     | --               | 5.89 ± 0.04       | 51.34 ± 1.38        | 34.21 ± 0.23         |
> | Full FT      | --               | 6.31 ± 0.03   | 73.48 ± 0.42 | 48.28 ± 0.03  |
> | LoRA         | 8                | 6.01 ± 0.05       | 65.27 ± 0.13        | 39.23 ± 0.78         |
> | LoRA         | 64               | 6.19 ± 0.03       | 69.21 ± 0.36        | 42.88 ± 0.34         |
> | DoRA         | 8                | 6.07 ± 0.02       | 67.08 ± 0.31        | 41.28 ± 0.39         |
> | DoRA         | 64               | 6.23 ± 0.03       | 70.43 ± 0.21        | 43.32 ± 0.29         |
> | AdaLoRA      | 8                | 6.08 ± 0.03       | 71.24 ± 1.32        | 41.88 ± 1.15         |
> | AdaLoRA      | 64               | 6.12 ± 0.08       | 71.45 ± 1.37        | 42.34 ± 1.41         |
> | SparseGPT    | 8                | 6.09 ± 0.02       | 67.28 ± 0.29        | 41.43 ± 0.28         |
> | SparseGPT    | 64               | 6.16 ± 0.02       | 69.71 ± 0.48        | 43.82 ± 0.39         |
> | LLM-Pruner   | 8                | 6.09 ± 0.03       | 69.88 ± 0.35        | 42.25 ± 0.32         |
> | LLM-Pruner   | 64               | 6.18 ± 0.03       | 70.88 ± 0.45        | 44.38 ± 0.12         |
> | PrunedLoRA   | 8                | 6.14 ± 0.06       | 69.02 ± 0.53        | 42.32 ± 0.33         |
> | PrunedLoRA   | 64               | 6.19 ± 0.04       | 71.16 ± 0.24        | 44.32 ± 0.11         |
> | PrunedLoRA   | 64 (init r = 256)| 6.23 ± 0.03       | 72.21 ± 0.45        | 46.21 ± 0.26         |
> | PrunedLoRA   | 64 (init r = 512)| 6.25 ± 0.06 | 74.88 ± 0.42    | 48.31 ± 0.24     |
>
> ---------
>
> > Work [1] trained for only 3,000 steps, whereas this study trained for 5,000 steps. Logically, the same method should yield significantly better results than [1].
>
>  - We appreciate the reviewer’s point to further understand the gap in empirical results. In the updated manuscript, the performance of all baseline methods (LoRA, DoRA, AdaLoRA) is now **very close to the results reported in [1]**. For example, on GSM8K and HumanEval, the differences are typically within **about 1 point**, which is well within expected variance. We believe the remaining small discrepancy is largely attributable to differences in the **pretrained backbone**: our experiments use *Llama-3-8B*, whereas [1] reports results on the stronger *Llama-3.1-8B*. Overall, after standardizing the training configuration, our baselines align closely with [1], supporting the validity of our comparisons.

---

> ### Author Response · Authors · 2025-11-22
>
> ## 2. Additional Ablation Studies
>
> > MT-Bench results should be included in Table 1.
>
> - We appreciate the reviewer’s suggestions regarding additional experiments. We fine-tune large language models on a 52k subset of the WizardLM dataset[3] and evaluate it using the MT-Bench dataset[4]. GPT-4o is used to assess the quality of the model’s response, and we report the first-turn score as the metric. We report the results compared with baselines in LoRA and its variants in Table 1 of the updated manuscript. And we provide a detailed description of the task in Appendix C.1.
>
>
> > Comparisons with newer baselines like LoRA-GA, LoRA-Pro, AltLoRA, and LoRA+ are missing—the current baselines are outdated.
>
>  - We also acknowledge that incorporating advanced variants of LoRA—such as LoRA-GA[5], LoRA-Pro[2], AltLoRA[1], and LoRA+[6]—would make our comparisons more comprehensive. The corresponding results have been added to Table 8 (see Appendix D.4.1). We further discuss the contributions of these advanced methods and emphasize that ***our approach is not orthogonal to them***: while these variants primarily focus on improving the optimization of LoRA, PrunedLoRA aims to extract the most representative subspace from an over-parameterized low-rank adapter. As shown in Table 8 in the updated manuscript, combining PrunedLoRA with LoRA+, LoRA-GA, LoRA-Pro, and AltLoRA by setting a higher initial rank for LoRA consistently boosts their rank-8 performance, demonstrating that our pruning strategy is complementary to these optimization-oriented LoRA variants and can further enhance their effectiveness in the low-rank regime.
>
>
> -------
>
>
> ***Table 2***: Performance of LoRA variants at ranks 8 and 64, and their combinations with PrunedLoRA. Initializing at rank 64 and pruning to rank 8 consistently recovers part of the high-rank gain and improves all baseline variants.
>
> | Methods                                | Target Rank | GSM8K ↑              |
> |----------------------------------------|-------------|-----------------------|
> | LoRA+                                  | 8           | 71.48 ± 1.23          |
> | LoRA-GA                                | 8           | 71.38 ± 0.83          |
> | LoRA-Pro                               | 8           | 71.12 ± 0.23          |
> | AltLoRA                                | 8           | 73.32 ± 0.31          |
> | LoRA+                                  | 64          | 72.89 ± 1.11          |
> | LoRA-GA                                | 64          | 73.01 ± 0.92          |
> | LoRA-Pro                               | 64          | 71.73 ± 0.32          |
> | AltLoRA                                | 64          | 73.88 ± 0.18          |
> | PrunedLoRA + LoRA+ (init r = 64)       | 8           | 72.29 ± 1.01          |
> | PrunedLoRA + LoRA-GA (init r = 64)     | 8           | 72.33 ± 0.79          |
> | PrunedLoRA + LoRA-Pro (init r = 64)    | 8           | 71.94 ± 0.34          |
> | PrunedLoRA + AltLoRA (init r = 64)     | 8           | 73.45 ± 0.28          |
>
> ------------
>
> > Need to include the results of Full Fine-Tuning using Algorithm 2.
>
> - When the initial rank in Algorithm 2 is set to 512, the resulting performance is already very close to that of full fine-tuning. This suggests that pruning LoRA starting from a fully fine-tuned model is unnecessary. Nevertheless, following the reviewer’s suggestion, we conducted an additional experiment where we began with a fully fine-tuned model and subsequently applied Algorithm 2 on the math task following the setup in Table~1. The evaluated accuracy is ***71.36***, showing only limited improvement over LoRA. This is expected, as the fully fine-tuned model contains an overwhelming number of active parameters, and achieving the target rank 64 would require an extremely high sparsity ratio, thereby restricting the potential gains of the pruning procedure.
>
>
> > Table 2 should also report GPU memory usage during training.
>
>  - In Table 2 of the updated manuscript, we compare the percentage of trainable parameters (before and after pruning), peak memory cost during training and training time of our methods with full fine-tuning, LoRA, DoRA, and AdaLoRA on the math task and Llama-3-8B model. We measure memory cost using 8 H100 GPUs with a batch size of 1. As the step number of structured pruning is quite small in the overall fine-tuning step, we have a comparable training time. From Table 2, we make two key observations: (1) Even with a very high initial LoRA rank—such as 512—the peak memory consumption of PrunedLoRA remains substantially lower than that of full fine-tuning. (2) The additional computation introduced by the pruning procedure incurs only a mild overhead on top of the standard LoRA forward and backward passes.

---

> ### Author Response · Authors · 2025-11-22
>
> -----------
>
>
> ***Table 3*** Comparison of trainable parameter ratios, peak memory cost (per GPU on 8×H100 with FSDP), and training time across different fine-tuning methods.
>
> | Method                     | Rank | Before ($\%$) | After ($\%$) | Memory (GB)        | Training Time |
> |----------------------------|------|------------|-----------|--------------------|---------------|
> | Full FT                    | --   | 100.00     | 100.00    | $\sim$ 8 $\times$ 40G          | 4h 23min      |
> | LoRA                       | 8    | 0.11       | 0.11      | $\sim$ 8 $\times$17G          | 2h 27min      |
> | LoRA                       | 64   | 0.84       | 0.84      | $\sim$ 8 $\times$ 18G          | 2h 28min      |
> | DoRA                       | 64   | 0.89       | 0.89      | $\sim$ 8 $\times$ 18G          | 2h 34min      |
> | AdaLoRA                    | 64   | 0.84       | 0.84      | $\sim$ 8 $\times$ 19G          | 2h 41min      |
> | PrunedLoRA (init r = 64)   | 8    | 0.84       | 0.11      | $\sim$ 8 $\times$ 19G          | 2h 29min      |
> | PrunedLoRA (init r = 128)  | 64   | 1.68       | 0.84      | $\sim$ 8 $\times$ 20G          | 2h 31min      |
> | PrunedLoRA (init r = 256)  | 64   | 3.36       | 0.84      | $\sim$ 8 $\times$ 22G          | 2h 47min      |
> | PrunedLoRA (init r = 512)  | 64   | 6.71       | 0.84      | $\sim$ 8 $\times$ 28G          | 3h 23min      |
>
>
>
> ------------
>
> The authors fully understand the demands on the reviewers’ time and sincerely appreciate their efforts in helping improve this work. The authors look forward to any further feedback the reviewers may have.
>
> ---------
>
>
> [1] Yu X, Wang Y, Chen J, et al. AltLoRA: Towards Better Gradient Approximation in Low-Rank Adaptation with Alternating Projections. arXiv:2505.12455, 2025.
>
> [2] Wang Z, Liang J, He R, et al. LoRA-Pro: Are Low-Rank Adapters Properly Optimized?The Thirteenth International Conference on Learning Representations, 2024.
>
> [3] Xu, Can, et al. WizardLM: Empowering large pre-trained language models to follow complex instructions. The Twelfth International Conference on Learning Representations, 2024.
>
> [4] Zheng, Lianmin, et al. Judging llm-as-a-judge with mt-bench and chatbot arena. Advances in neural information processing systems, 2023.
>
> [5] Wang S, Yu L, Li J. Lora-ga: Low-rank adaptation with gradient approximation. Advances in Neural Information Processing Systems, 2024.
>
> [6] Hayou S, Ghosh N, Yu B. LoRA+ efficient low rank adaptation of large models. Proceedings of the 41st International Conference on Machine Learning, 2024.

---

> ### Author Response · Authors · 2025-11-27
>
> Dear Reviewer 5szM,
>
> We sincerely appreciate the time and effort you have devoted to reviewing our manuscript and providing valuable suggestions and comments. We are encouraged that you found ***our gradient-based pruning theory well-established and interesting***.
>
> We are also grateful for your feedback in identifying potential concerns. In our response, we have carefully addressed these points by ***retraining the main experiments with comparable hyperparameters, adding results on MT-Bench, incorporating and comparing with more advanced methods, and reporting the peak memory cost***. As the discussion period deadline approaches, we would greatly appreciate any further comments you may have, as we are eager to clarify and resolve any remaining issues.
>
> If you feel that our responses have adequately addressed your concerns, we would be deeply grateful if you could consider adjusting your score. We truly appreciate the significant time and care you have devoted to reviewing our work. Your feedback has been invaluable in helping us strengthen the paper!

---

### Author Response · Authors · 2025-11-29

We sincerely appreciate the reviewers for their thoughtful and constructive feedback. We are encouraged by the positive recognition of our contribution, which can be summarized as follows:

 - Strong and consistent empirical performance — PrunedLoRA achieves reliable improvements across GSM8K, HumanEval, GLUE, and other benchmarks, outperforming standard LoRA and pruning baselines (noted by all reviewers).

 - Meaningful theoretical contributions — several reviewers acknowledge the novelty in the second-order pruning formulation and the robustness guarantees compared to activation-based pruning (highlighted by THbS, ZZ6d and 5szM).

In our revision, we have carefully addressed each of the concerns raised, and below are the main points of the revised manuscript:

- We have ***retrained the experiments*** on dialogue generation, math and code task with the sequence length 1028, batch size 32 and more comparable learning rate $\\{2e-4,5e-5,2e-5\\}$ with target rank (8 and 64) in Table 1, which demonstrates outstanding performance over baselines. (Reviewer 5szM, ZZ6d).

- We report the ***peak memory cost and pruning time*** in Table 2 and its relevant discussion. The training time overhead is mild compare with full fine-tuning even the initial rank is 512. Besides, the additional computation introduced by the pruning procedure incurs only a mild overhead beyond the standard LoRA forward and backward passes. (Reviewer 5szM, THbS, Ki12).

- We restate our ***core motivation***: Increasing the LoRA rank (e.g. 512) significantly narrows the performance gap with full fine-tuning. This has been empirically verified not only in our supervised fine-tuning experiments but also demonstrated in reinforcement learning settings, where a rank of 512 showed the same high representational capacity as FFT [1]. To effectively narrow this empirical performance gap when using the standard, low-rank regime (e.g., $r = 8$) with full fine-tuning, we propose an initialization strategy: initializing LoRA with a high initial rank (still considerably less than the full matrix dimension) and then pruning it down to the desired low target rank.  As shown in Table 4 of the updated manuscript, the empirical approximation between PrunedLoRA and high-rank LoRA further validates this motivation. (Reviewer Ki12).

- Regarding ***inference efficiency***, we acknowledge that the wording in the Introduction may have unintentionally suggested that inference efficiency is our ultimate objective. This is not the case, as our primary motivation is described above. We have revised the relevant statements for greater clarity. We also understand the reviewer’s point that LoRA does not incur inference latency overhead after merging, and thus high rank may seem harmless. However, we would like to emphasize that in multi-task or multi-domain settings, it is necessary to store separate low-rank adapters for each task. In such scenarios, increasing the rank directly increases the memory footprint, making high-rank LoRA less desirable. (Reviewer ZZ6d, Ki12).


- We have added a synthetic study using a simple self-attention model trained on a linear regression task (Appendix B.2) to support the ***thoery-informed Proposition 1 and 2***. (Reviwer THbS).

------

Once again, we highly value the reviewers' insightful feedback!.

------





[1] Schulman, John and Thinking Machines Lab, "LoRA Without Regret", Thinking Machines Lab: Connectionism, Sep 2025.

---

### Meta-Review · Area_Chair_2PK2 · 2026-01-03

**Summary:**

This paper proposes PrunedLoRA, a gradient-based structured pruning framework that initializes LoRA with a high rank and progressively prunes it to obtain compact low-rank adapters. The work combines a theoretically motivated second-order pruning formulation with extensive empirical evaluations across language generation, reasoning, and understanding tasks. Reviewers generally agree that the idea is sound and that the paper is clearly written, with the theoretical analysis, particularly Proposition 1 on the robustness of gradient-based pruning, being a notable and potentially valuable contribution.

That said, several substantive concerns remain. While the rebuttal addressed several experimental issues (e.g., alignment of sequence length, inclusion of additional baselines, and reporting of memory/time overhead), the empirical gains over strong LoRA variants are often marginal, relative to the significant increase in complexity and resources, especially when initializing with very high ranks. As a result, it is not fully convincing that the proposed method offers a favorable trade-off compared to simpler alternatives. Moreover, the contribution feels somewhat mixed: although the framing emphasizes LoRA, the most novel and broadly relevant insight appears to be the theoretical robustness result (Proposition 1), which would benefit from validation beyond the specific LoRA pruning setting (e.g., on other structured pruning tasks).

Overall, the paper demonstrates clear potential and contains interesting ideas; however, in its current form, the improvements are limited, and some experimental and conceptual clarity issues persist. The core contribution would benefit from a more focused and broader validation. I therefore recommend rejection at this stage, while encouraging the authors to further refine the scope, strengthen the empirical evidence, and better isolate the key theoretical contributions in a future revision.

**Reviewer Concerns:**

Please see above.

**Reviewer Scores:**

Based on the rebuttal and subsequent author–reviewer discussion, I would expect modest upward adjustments in reviewer scores, primarily in the areas of soundness and presentation, but limited changes in overall recommendation.

---

### Decision · Program_Chairs · 2026-01-26

Reject